# Bumblebee visual allometry results in locally improved resolution and globally improved sensitivity

Gavin J Taylor[1]*, Pierre Tichit[1], Marie D Schmidt[1,2], Andrew J Bodey[3], Christoph Rau[3], Emily Baird[1,4]*

[1]Department of Biology, Lund University, Lund, Sweden; [2]Westphalian University of Applied Sciences, Bocholt, Germany; [3]Diamond Light Source, Oxfordshire, United Kingdom; [4]Department of Zoology, Stockholm University, Stockholm, Sweden

**Abstract** The quality of visual information that is available to an animal is limited by the size of its eyes. Differences in eye size can be observed even between closely related individuals, yet we understand little about how this affects vision. Insects are good models for exploring the effects of size on visual systems because many insect species exhibit size polymorphism. Previous work has been limited by difficulties in determining the 3D structure of eyes. We have developed a novel method based on x-ray microtomography to measure the 3D structure of insect eyes and to calculate predictions of their visual capabilities. We used our method to investigate visual allometry in the bumblebee *Bombus terrestris* and found that size affects specific aspects of vision, including binocular overlap, optical sensitivity, and dorsofrontal visual resolution. This reveals that differential scaling between eye areas provides flexibility that improves the visual capabilities of larger bumblebees.

DOI: https://doi.org/10.7554/eLife.40613.001

**\*For correspondence:**
gavin.taylor.01@gmail.com (GJT);
emily.baird@zoologi.su.se (EB)

**Competing interests:** The authors declare that no competing interests exist.

## Introduction

What an animal can see within its environment is restricted by its visual field, or the total angular region of the world from which light can be absorbed by its photoreceptors. To detect specific objects within this visual field, the eyes need spatial resolution, which is achieved through (and limited by) the arrangement of individual receptors that sample the spatial distribution of light (*Land and Nilsson, 2012*). Having an eye with spatial resolution allows an animal to detect differences in the intensity of the light reaching it from different directions, and this information is crucial for the myriad of visually guided behaviours exhibited by different species (*Cronin et al., 2014*).

Over a given finite area, an eye cannot maximise resolution without sacrificing sensitivity (the amount of light captured) (*Land, 1997*) – increased resolution requires light to be sampled from a decreased region of space, necessarily reducing sensitivity (*Snyder, 1979*). As a result, the relative density and optical properties of receptors often vary topologically across an eye, creating variations in visual resolution and optical sensitivity that 'fine-tune' the capabilities in certain regions of the visual field. We are familiar with this from our own eyes – our fovea provides high-resolution vision over a small region of space, while our peripheral vision is blurred but views a much larger portion of the world. The eyes of other species have also evolved specializations that enable them to acquire critical information from important regions of the world, such as elongated regions of acute vision for detecting the horizon (*Dahmen, 1991*) and 'bright zones' of high optical sensitivity for discriminating passing prey or potential mates against a bright background (*Straw et al., 2006*). Specialized areas represent a local investment in improving a specific visual capability that is related to an animal's behavioural and ecological requirements. Determining the topology of visual capabilities

**eLife digest** Bees fly through complex environments in search of nectar from flowers. They are aided in this quest by excellent eyesight. Scientists have extensively studied the eyesight of honeybees to learn more about how such tiny eyes work and how they process and learn visual information. Less is known about the honeybee's larger cousins, the bumblebees, which are also important pollinators. Bumblebees come in different sizes and one question scientists have is how eye size affects vision.

Bigger bumblebees are known to have bigger eyes, and bigger eyes are usually better. But which aspects of vision are improved in larger eyes is not clear. For example, does the size of a bee's eyes affect how large their field of view is, or how sensitive they are to light? Or does it impact their visual acuity, a measurement of the smallest objects the eye can see? Scaling up an eye would likely improve all these aspects of sight slightly, but changes in a small area of the eye might more drastically improve some parts of vision.

Now, Taylor et al. show that larger bumblebees with bigger eyes have better vision than their smaller counterparts. In the experiments, a technique called microtomography was used to measure the 3D structure of bumblebee eyes. The measurements were then applied to build 3D models of the bumblebee eyes, and computational geometry was used to calculate the sensitivity, acuity, and viewing direction across the entire surface of each model eye. Taylor et al. found that larger bees had improved ability to see small objects in front or slightly above them. They had a bigger area of overlap between the sight in both eyes when they looked forward and up. They were also more sensitive to light across the eye. The experiments show that improvements in eyesight with larger size are very specific and likely help larger bees to adapt to their environment.

Behavioral studies could help scientists better understand how these changes help bigger bees and how the traits evolved. These findings might also help engineers trying to design miniature cameras to help small, flying autonomous vehicles navigate. Bees fly through complex environments and face challenges similar to those small flying vehicles would face. Emulating the design of bee eyes and how they change with size might lead to the development of better cameras for these vehicles.

DOI: https://doi.org/10.7554/eLife.40613.002

across an eye's entire field of view (FOV) can provide important insights into the visually guided behaviours and environment of its owner (*Moore et al., 2017*) – for example, the topology of facet size (either with or without a region of enlarged facets) on the eyes of male bumblebees indicates their species-preferred mating strategy (perching or patroling, respectively) (*Streinzer and Spaethe, 2014*).

Across a wide range of animal groups, bigger individuals generally have absolutely larger eyes, although eyes do not typically grow linearly with body size but, rather, become proportionally smaller in larger animals (*Jander and Jander, 2002*; *Howland et al., 2004*), even within a species (*Perl and Niven, 2016a*). Increasing eye size allows improvements in visual quality; in a bigger eye, resolution can be increased by adding receptors and optical sensitivity can be increased by enlarging the receptor size. The relationship between the growth of any trait and an animal's total body size is conventionally modeled using a power function ($Y = bx^{\alpha}$) to relate a trait ($Y$) to a measure of body size ($x$). This provides two parameters that describe allometry, the scaling exponent ($\alpha$) and the initial growth index ($b$) *Huxley and Teissier (1936)*, where anatomical features that increase in size at a slower rate than the body are represented by an exponent less than 1. Scaling exponents for eye size within invertebrate groups are usually higher than those of vertebrates (which average 0.6) (*Howland et al., 2004*) and even include an unusual positive allometry rate in stingless Meliponine bees (*Streinzer et al., 2016*). Allometry studies within hymenopteran species – in which body size can vary substantially between conspecifics – have shown that, although larger individuals of several ant species primarily invest in increasing their total facet number (*Klotz et al., 1992*; *Zollikofer et al., 1995*; *Schwarz et al., 2011*), other ants (*Baker and Ma, 2006*; *Perl and Niven, 2016a*) and bumblebees (*Kapustjanskij et al., 2007*) increase both the number of facets and their size. Allometry has even been shown to vary across wood ant eyes, in which differential scaling

exponents of facet size – leading to differences in visual capabilities – are found between eye areas (*Perl and Niven, 2016b*). These insect species all possess apposition compound eyes (in which each lens focuses light onto an individual receptor) that, to avoid losing sensitivity, must grow in proportion to the square of a resolution improvement, as both the size and number of lenses must be increased (*Land and Nilsson, 2012*). Given that homogeneously increasing the size of a compound eye provides a relatively small improvement in its overall visual capability, we hypothesise that compound eyes are likely to scale non-uniformly, such that the majority of a larger eye will be invested in improving vision in a small portion of the FOV.

To test whether increasing compound eye size does indeed lead to the development or improvement of specialised visual regions in larger individuals, it is necessary to link the allometry of eye properties to the visual capabilities they provide. Visual resolution in compound eyes is often estimated by dividing an assumed hemispherical FOV (*Land, 1997*) by the number of facets, leading directly to the conclusion that resolution has the same (although negative) scaling exponent as facet number (*Jander and Jander, 2002*). This assumption is not supported, however, by direct measurements of inter-ommatidial (IO) angle (a measure of local visual resolution). For instance, IO angles from both desert ant and fruit fly eyes have absolute scaling exponents (−0.40 and −0.21, respectively) that are lower than those for the facet number (0.75 and 0.58) (*Zollikofer et al., 1995*; *Currea et al., 2018*). In addition, the scaling exponent of IO angle was found to vary between different regions of Orange Sulphur butterfly eyes, whereas the exponent of facet diameter remained relatively constant across the eye (*Merry et al., 2006*). A complication when comparing corneal topology between different compound eyes is that eye shape can vary substantially, both between groups (for example, butterflies have nearly hemispherical eyes (*Rutowski, 2000*), whereas flatter oval eyes are common in hymenopterans (*Jander and Jander, 2002*)) and within groups (for example, male and female honeybees have drastically different eye shapes (*Streinzer et al., 2013*)). In the absence of a common reference frame, topologies based on general descriptions of eye shapes (for example, the well-known dorsal rim area (*Labhart and Meyer, 1999*)) are not necessarily suitable for comparing visual capabilities between or within species, because corresponding anatomical areas may ultimately have different fields of view. Linking points on the eye to their projection into the visual world – and defining the visual capabilities that they provide – is necessary not only for comparing vision between species but also for gaining an understanding of how an eye samples information from the environment and how this influences the control of visually guided behaviours, such as foraging, predation, and mating.

We have begun to explore the effect of size on visual capacity by comparing the topology across the entire eyes of individuals of the size-polymorphic bumblebee *Bombus terrestris*. The size differences within this species – which can equal an order of magnitude in mass (*Goulson, 2003*) – influence several of its visually guided behaviours (*Spaethe and Weidenmüller, 2002*; *Spaethe and Chittka, 2003*). Yet, little is known about how optical sensitivity and visual resolution across a bumblebee's entire visual field are affected by its eye size. To test our hypothesis *that the differential scaling of compound eyes will primarily improve the visual capabilities in only a small region of the visual field*, we developed a novel method based on constructing 3D models of apposition compound eyes that were imaged with x-ray microtomography (microCT, *Figure 1A*) (*Baird and Taylor, 2017*). These models allowed us to approximate the FOV of an eye from its corneal projection (CP) (*Figure 1E*), over which the facet topology was mapped (*Figure 1C*). Using this method, we directly calculated the inter-facet (IF) angle (as an approximation of IO angle) across the world, as well as the projected topologies of eye properties affecting optical sensitivity – namely, the facet diameter and the retinal and lens thicknesses (*Figure 1D*). Crucially, this technique allowed us to place eye properties and visual capabilities from different eyes into common, world-referenced coordinates. We utilized our technique to investigate how size polymorphism influences *B. terrestris* vision by calculating the topology of eye properties and the visual capabilities of six worker bees varying in linear body size by ×2.8 and in eye volume (EV) by ×3.9. As a reference, we also analyzed the eyes of European honeybees (*Apis mellifera*), a species that has highly consistent worker size. Unlike those of *B. terrestris*, the eyes of honeybees have been the subject of extensive behavioural, anatomical, and physiological analyzes, and provide important data against which the results obtained from our method can be critically compared. Moreover, we identified key features in the allometry of the visual topology of bumblebees that may represent general rules for how compound eyes scale with size.

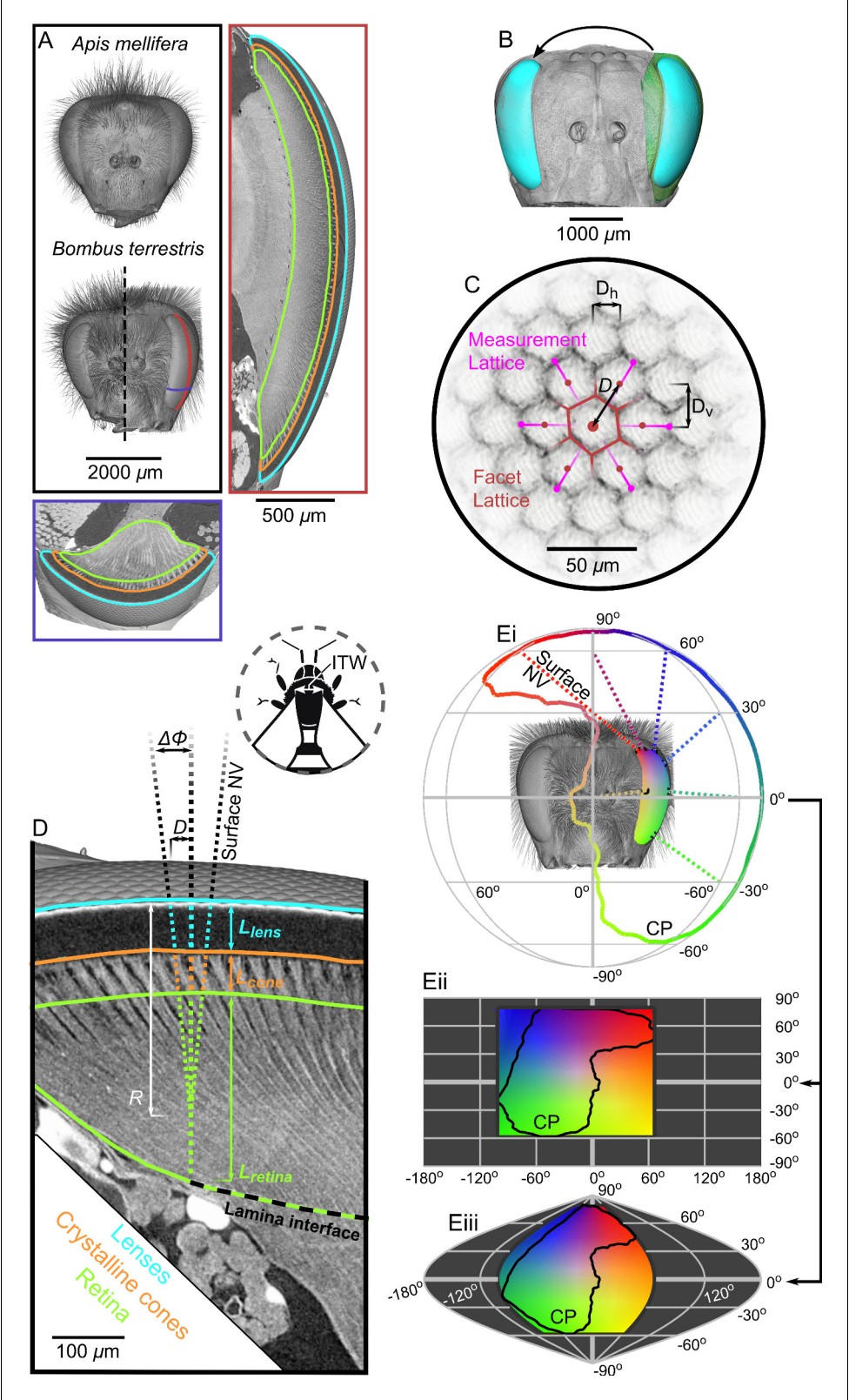

**Figure 1.** Tomographic images of bee eyes and visual analysis. (**A**) Volume rendering of the dried heads of workers from the bee species used in this study. The right-hand side of a small *B. terrestris* head is shown (intertegular width (ITW) = 3.0 mm) in comparison to left hand side of a large individual from the same colony (ITW = 5.4 mm, note that the mandibles and some hair of this bee were too large to image). The ITW of the *A. mellifera* specimen was 3.6 mm. The box on the right displays a vertical section along the midline of a *B. terrestris* apposition compound eye showing the gross

*Figure 1 continued on next page*

*Figure 1 continued*

morphology of the lenses, crystalline cones (CC), and retina (indicated using colored outlines as in panel **D**). A portion of the optic lobe is also visible to the left of the retina. The lower box shows a transverse section across the ventral portion of the compound eye, showing the same features as in the vertical section. The approximate location of the sections are indicated with lines on the larger *B. terrestris* eye; both sections have the same scaling. (**B**) A volume rendering of the left compound eye (green) of each bee was aligned onto a rendering of the full head of another bee (grey), which allowed the segmented eye (cyan) to be placed relative to the head and also mirrored to the right side. (**C**) To measure the local facet dimensions, six points were selected on the opposing borders of the six facets that surrounded a central facet. These points were then used to compute the local structure of the facet lattice from which the lens diameter $D$ was calculated (the horizontal ($D_h$) and vertical ($D_v$) facet dimensions were not used in this study). (**D**) Surface normal vectors (NV) were calculated from the exterior surface of the lenses to indicate the local viewing direction. The IF angle ($\Delta\Phi$) was defined as the difference between viewing directions at a distance of one lens diameter. The NV was traced into the eye and the points where it intersected the front surface of the CC, the retina, and the lamina interface were used to determine the thickness of these structures ($L_{lens}$, $L_{cone}$, and $L_{retina}$, respectively). Note that the angle of the CC and the photoreceptors in the retina can be misaligned from the corneal NV, in which case the IF can differ from the IO angle. Inset, a diagram shows the measurement of ITW as the distance between the wing bases on a bee's thorax. (**Ei**) The projection of the NVs (several are plotted as dotted lines) from the eye onto a sphere indicates the extent of the eye's corneal projection (CP), which can also be represented on equirectangular (**ii**) or sinusoidal (**iii**) projections on which the CP is indicated by black lines. The color coding indicates the viewing direction, and was applied by stretching a 2D color map across the CP extent on the equirectangular projection in (**ii**); equivalent colors indicate the same viewing direction in each projection.

DOI: https://doi.org/10.7554/eLife.40613.003

## Results

Bumblebee eyes display negative allometry and increase in both area and volume at a slower rate than their body size (*Figure 2A*, *Supplementary file 1*–Table S1). Their corneal projection (CP) – the angular projection of the cornea onto the world – also increases with eye size (*Figure 2C*), although at a slower rate than the total number of facets (*Figure 2B*). Naively, this suggests that bigger bumblebees would have more facets per unit area, implying smaller IF angles. However, our calculations suggest that IF angles generally maintain a relatively similar, although broad, distribution as eye size

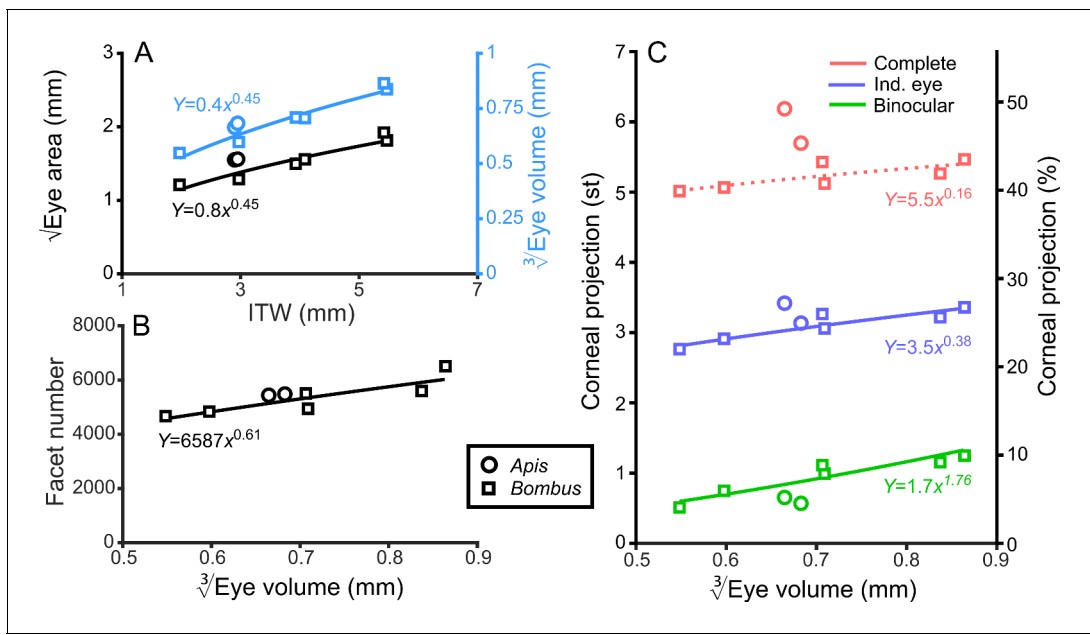

**Figure 2.** General analysis of differently sized compound eyes. (**A**) $\sqrt{\text{Eye area}}$ and $\sqrt[3]{\text{EV}}$ (on the right-hand Y-axis) vs. ITW. (**B**) The number of facets in each compound eye as a function of $\sqrt[3]{\text{EV}}$. (**C**) The size of the individual (Ind.), binocular, and complete CPs both in steradians (left Y-axis, max.: $4\pi$) and as a percentage of the total visual sphere (right Y-axis, max.: 100%). Squares denote bumblebees whereas circles denote honeybees in all plots, while color coding is explained on each panel. A power function was fitted to the *Bombus* measurements for each parameter (*Supplementary file 1*–Table S1); the resulting functions are written and plotted in each panel. The dotted line in panel C indicates that the correlation between the complete CP and $\sqrt[3]{\text{EV}}$ was not significant.

DOI: https://doi.org/10.7554/eLife.40613.004

increases, with only a small decrease in the mean IF angle present in the largest bees. The smallest IF angles were slightly less than 1° for all bees (*Figure 3A*). Our facet-based average for IF angles is between the values generated by the hemispherical assumption (*Figure 3A*, triangles) and what would be predicted by dividing the calculated CP by the number of facets (*Figure 3A*, stars). Nonetheless, the results of both calculations based on facet number do lie within the range of values observed for each bee (*Figure 3A*). The IF angle (ΔΦ) is itself derived from local eye properties: the facet diameter (D) divided by the radius of curvature (R). When considering the allometry of these eye properties, we found that both facet diameter (*Figure 3C*) and radius (*Figure 3—figure supplement 1A*) clearly increase with eye size. As both properties had a similar scaling exponent (*Supplementary file 1*–Table S1), this results in similar IF angles across the range of *Bombus* eye sizes examined.

## Fields of view

The CP increases with bumblebee eye size (*Figure 2C*), but it appears that this increase does not result from simply enlarging the CP in all directions (*Figure 4A,B*). The dorsolateral limit of the CP of each eye is relatively consistent (between 30° to 60° elevation (el.) and −90° to −60° azimuth (az. - Azimuth), *Figure 4B*), while bigger bees appear to enlarge their CP dorsofrontally (between 0° to 90° el. and −15° to 75° (az. - Azimuth)) and to a lesser extent ventrolaterally (between 0° to −60° el. and −105° to −60° (az. - Azimuth)). When the CP of each bee's right eye is also considered, it is apparent that all bees have regions of binocular CP overlap (*Figure 4C*) that increase in angular area with eye size (*Figure 2C*). Increasing binocularity is primarily observed dorsofrontally (between 30° to 90° el. and −75° to 75° (az. - Azimuth), *Figure 4C*), but a wedge of binocularity is also observed facing directly forwards (between −30° to 30° el. and −15° to 15° (az. - Azimuth)). Given the shapes of these areas, these appear to be two distinct binocular regions; they are separated on the smallest bee and merge as the binocular field increases in angular size in bigger bees. The CP is a spatial, but binary, representation of each bee's field of vision across which the eye properties and visual capabilities vary topologically.

## Optical sensitivity

Optical sensitivity is influenced by facet diameter and rhabdom length (and also by acceptance angle) (*Warrant and Nilsson, 1998*) and these properties increase with eye size (*Figure 3C and 5A*). This suggests that larger bees have more sensitive ommatidia than smaller bees if the acceptance angles of receptors are assumed to be equivalent to the calculated IF angles and the retinal thickness is used as a proxy for rhabdom length (*Figure 5—figure supplement 1*). Taking into consideration the similar eye-wide averages for IF angle (*Figure 3A*), an increase in retinal thickness and facet diameters suggests that bigger bees invest in improving their sensitivity rather than their visual resolution. There is, however, substantial variation in the histograms of the measured variables for each eye (*Figure 3A,C* and *5A*), so it will be interesting to investigate the topology of how these are projected into the visual field.

## Projected topologies and profiles

The smallest IF angles within each bee's CP are observed in a laterally positioned vertical band running from −45° until 60° el. at approximately −60° (az. - Azimuth), whereas the greatest IF angles are observed at the posterior and rightmost dorsal limits of the visual field (*Figure 3B*). All *Bombus* have a similar average IF angle profile across elevation (average ~1.5° between −30° to 30° el., *Figure 3Bi*). When averaged across azimuth, it is apparent that larger eyes have greater frontal resolution (between −45° to 15° (az. - Azimuth), *Figure 3Bii*), although the lowest azimuthal average IF angle is *not* directed frontally, but rather at ~−60° for all bees. A distinctly different topology is present when projecting the facet diameters into each bee's CP (*Figure 3D*). Larger *Bombus* individuals have larger facets (*Figure 3C*), and each bee's facet diameters (averaged across elevation) are greatest ventrally (<0° el., *Figure 3Di*) and smallest dorsally (>45° el.). The facet diameters of each bee are generally similar within these elevation ranges, but they are connected by a transitory range as diameters decrease from an elevation of 0° to 45°. When averaging facet diameter across azimuth, each *Bombus* has its largest facets facing laterally (<60° (az. - Azimuth), *Figure 3Dii*). Retinal thickness is projected into visual space with a similar, although less consistent, topology as

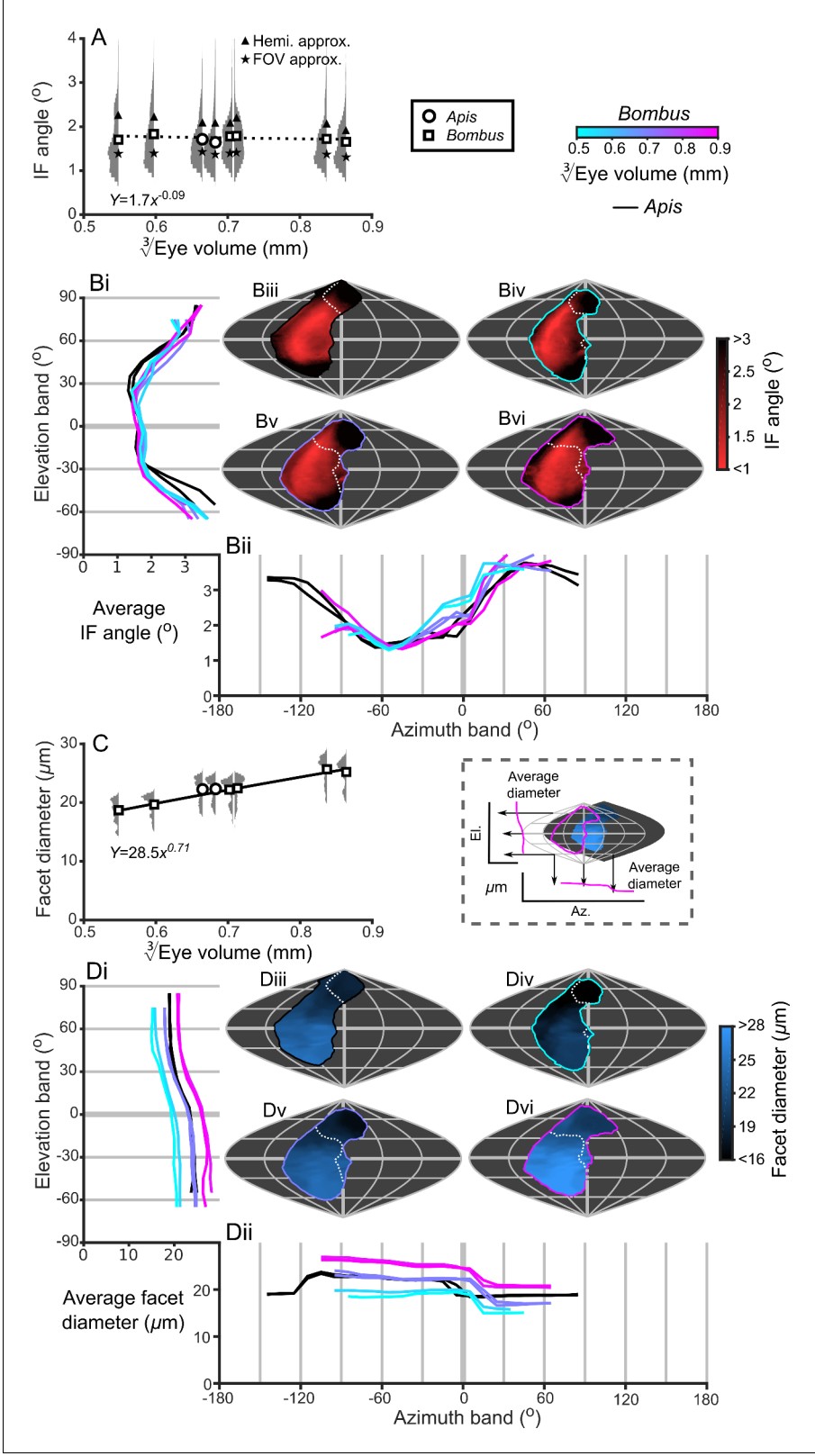

**Figure 3.** IF angle and facet diameter averages and projections. (A) Average IF angles and relative distribution of IF angles for each bee. The triangles indicate the results of dividing a hemisphere by the total facet number to predict the average IF angle (Hemi. approx.) (***Land, 1997***), while the stars indicate the results of dividing an individual eye's angular CP by its total facet number (FOV approx.). (B) The topographic distribution of IF angle

*Figure 3 continued on next page*

*Figure 3 continued*

(indicated by the red-black color gradient bar) projected from each bee's eye, shown as sinusoidal projections for a honeybee (iii) and for small- (iv), medium- (v) and large-sized (vi) bumblebees (azimuth and elevation lines are plotted at 60° and 30° intervals, respectively). Profiles of the average IF angle are shown for elevation (panel Bi, where IF angle is averaged across *all* azimuth points in each eye's CP) and azimuth (panel Bii, where IF angle is averaged across *all* elevation points in each eye's CP). The diagram below panel Bii provides a graphical representation of averaging topologies to profiles. (C) Averages and relative distributions of the facet diameters from each bee (bees as in panel A). (D) As for panel **B** but showing the topology (indicated by the blue-black color gradient bar) and profiles of average facet diameter. Squares denote bumblebees and circles denote honeybees in panels **A and C**, and a power function was fitted to the *Bombus* measurements for the parameters in these panels (*Supplementary file 1*–Table S1); the resulting functions are written and plotted in each panel. The dotted line in panel A indicates that the correlation between the IF angle and $\sqrt[3]{\mathrm{EV}}$ was not significant. The cyan-to-magenta color bar indicates the $\sqrt[3]{\mathrm{EV}}$ of the *Bombus* profiles in panels **B** and **D**, while black lines indicate honeybees. The white dotted lines across the topologies in panels B and D (iii to vi) indicate the border of a bee's binocular CP. See *Figure 3—figure supplements 1–4* for equivalent data on curvature, lens and crystalline cone thickness, and eye parameter. Correlations between each locally measured variable are plotted in *Figure 3—figure supplement 5*.

DOI: https://doi.org/10.7554/eLife.40613.005

The following figure supplements are available for figure 3:

**Figure supplement 1.** Description of the local radius of the curvature of compound eyes.

DOI: https://doi.org/10.7554/eLife.40613.006

**Figure supplement 2.** Description of the local lens thickness of compound eyes.

DOI: https://doi.org/10.7554/eLife.40613.007

**Figure supplement 3.** Description of the local CC thickness of compound eyes.

DOI: https://doi.org/10.7554/eLife.40613.008

**Figure supplement 4.** Description of the local eye parameter of compound eyes.

DOI: https://doi.org/10.7554/eLife.40613.009

**Figure supplement 5.** Correlation of local variable values.

DOI: https://doi.org/10.7554/eLife.40613.010

facet diameter; average thickness generally increases towards the ventral and lateral CPs (*Figure 5B*). Interestingly, lens thickness has a different topology to the other properties that we have described, as it peaks in the frontal visual field (*Figure 3*; *Figure 3—figure supplement 2Bii*) but is also correlated with facet diameter ($r = 0.65$, *Figure 3—figure supplement 5*). After examining the projection of these variables into visual space, it initially appears that bumblebee eye properties maintain a similar topology that essentially scales with eye size (facet diameter (*Figure 3D*), radius of curvature (*Figure 3—figure supplement 1B*), retinal thickness (*Figure 5B*), lens thickness (*Figure 3—figure supplement 2B*), and CC thickness (*Figure 3—figure supplement 3B*). Although the IF angle is a function of local facet diameter and radius of curvature, local variations in the IF angles (*Figure 3B*) evidently arise from subtle changes in eye properties.

## Mapping scaling rates

Because the projected topologies varied locally in shape (e.g. IF angle, *Figure 3B*), we calculated maps of the local scaling exponents for eye properties and visual capabilities in world-referenced coordinates to examine the spatial variation in the allometry of these characteristics. As identified from the IF angle profiles (*Figure 3B*), scaling exponent maps of IF angle also show that larger bees had improved resolution in their frontal and dorsofrontal CPs (between −15° to 60° el. and −30° to 10° (az. - Azimuth), *Figure 6A*). However, as the facet diameter maintains an almost uniform exponent across the visual field (*Figure 6B*), local variations in the scaling rate of the radius of curvature (*Figure 6—figure supplement 1C*) must cause differences in the IF angle exponent across the eye. In larger bumblebees, changes in the eye's local radius, and consequently its shape, are the primary determinant of changes in IF angle and, indeed, local IF angle has a strong negative correlation to the radius of curvature ($r = −0.84$) but is not correlated to facet diameter ($r = −0.07$, *Figure 3—figure supplement 5*). Both retinal thickness and facet diameter contribute to sensitivity and have positive scaling exponents, although the local exponent of retinal thickness shows substantially more variation (*Figure 6C*): it is highest in the dorsal hemisphere and lowest in the frontal region

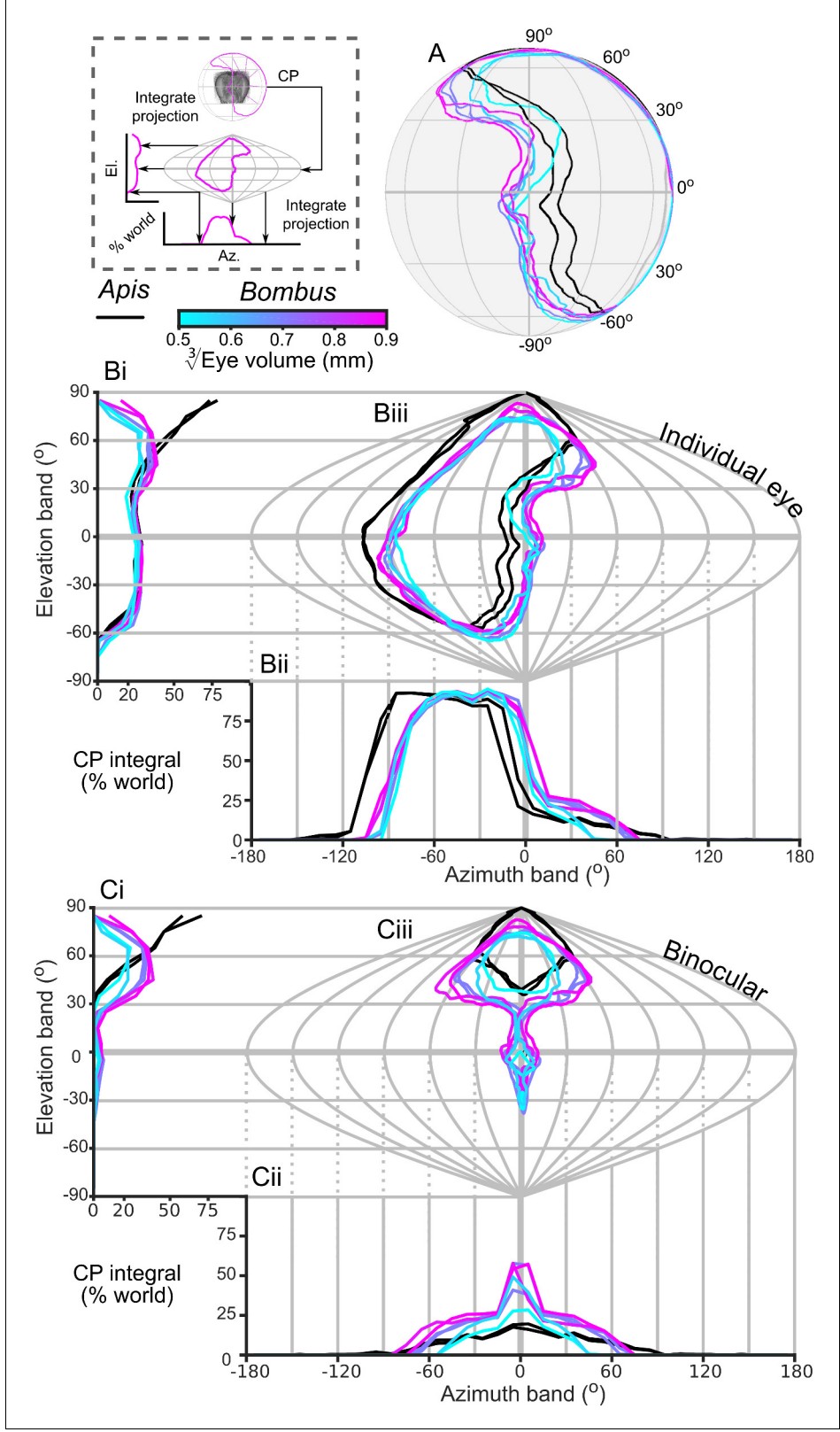

**Figure 4.** The corneal projection (CP) of compound eyes. (**A**) The CP of each bee's left eye shown on a sphere representing the world. (**B**) A sinusoidal projection and analysis of the CPs (iii). Profiles of integrated CP are shown across elevation (panel Bi, the integral of all azimuth points in the CP as a function of elevation) and azimuth (panel Bii, the integral of all elevation points in the CP as a function of azimuth) and are expressed as a percentage

*Figure 4 continued on next page*

*Figure 4 continued*

of the total number of points. (**C**) As for panel **B** but depicting the limit of the binocular overlap between the visual field of the left and right eyes for each bee. Colouring of the CP and profile lines is indicated by the color gradient bar as described in the caption of *Figure 3*. See *Figure 1E* for a visualization of the relationship between a sphere and its projections and the diagram to the left of panel A for a graphical representation of integrating CPs to profiles.

DOI: https://doi.org/10.7554/eLife.40613.011

associated with binocularity (*Figure 4C*). Arguably, maps of scaling exponents provide a clearer

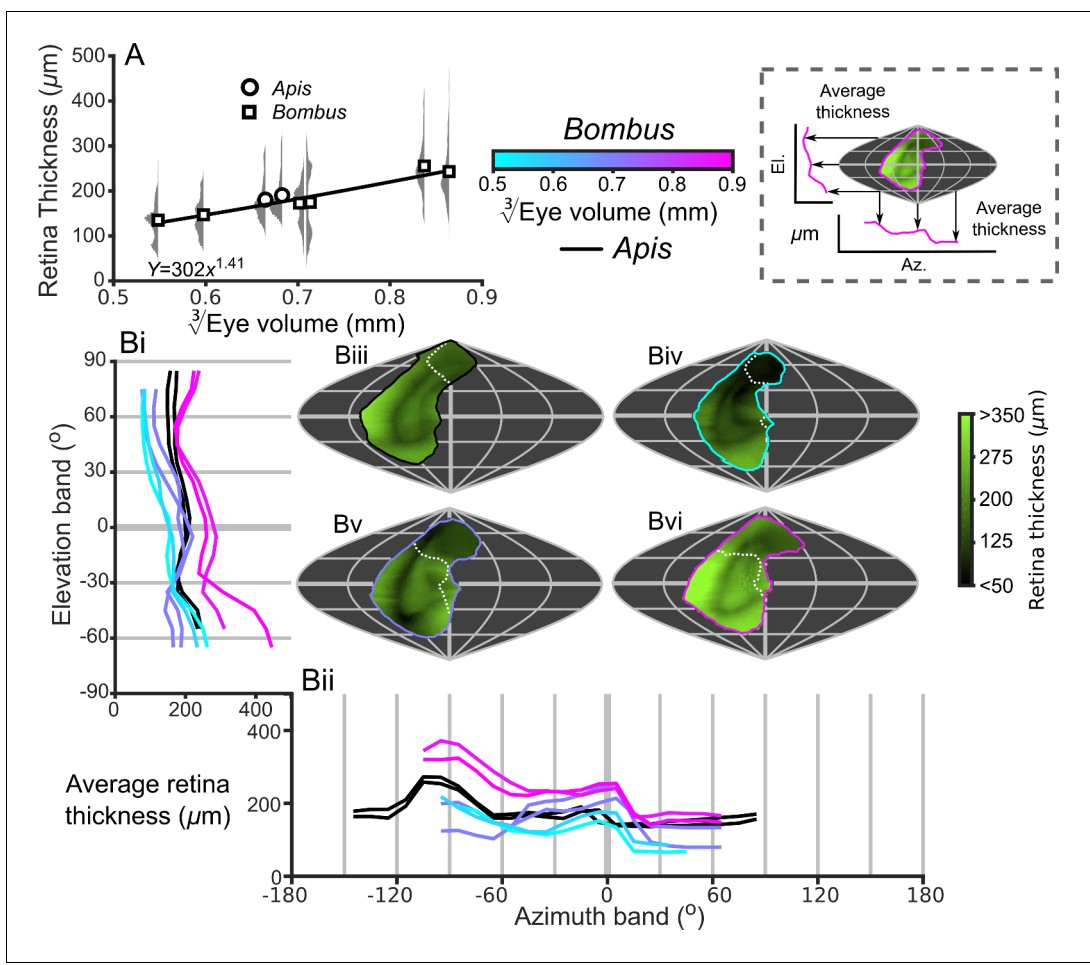

**Figure 5.** Description of the retinal thickness underlying compound eyes. (**A**) The average retinal thicknesses and relative distribution for each bee as a function of $\sqrt[3]{\mathrm{EV}}$ (notes in the caption for *Figure 2* also apply to this panel). (**B**) The topographic distribution of retinal thickness (indicated by the green-black color gradient bar) projected from each bee into the visual world, shown as sinusoidal projections for a honeybee (iii), and for small- (iv), medium- (v), and large-sized (vi) bumblebees. Profiles of the average retinal thickness are shown for elevation (panel Bi, where thickness is averaged as a function of elevation, as described in the caption for *Figure 3B*) and azimuth (panel Bii, thickness is averaged as a function of azimuth). Further details about projections and profiles are described in the caption of *Figure 3*. The average calculated sensitivity for each eye is show in *Figure 5— figure supplement 1*.

DOI: https://doi.org/10.7554/eLife.40613.012

The following figure supplement is available for figure 5:

**Figure supplement 1.** The average optical sensitivity calculated for each bee vs $\sqrt[3]{\mathrm{EV}}$.

DOI: https://doi.org/10.7554/eLife.40613.013

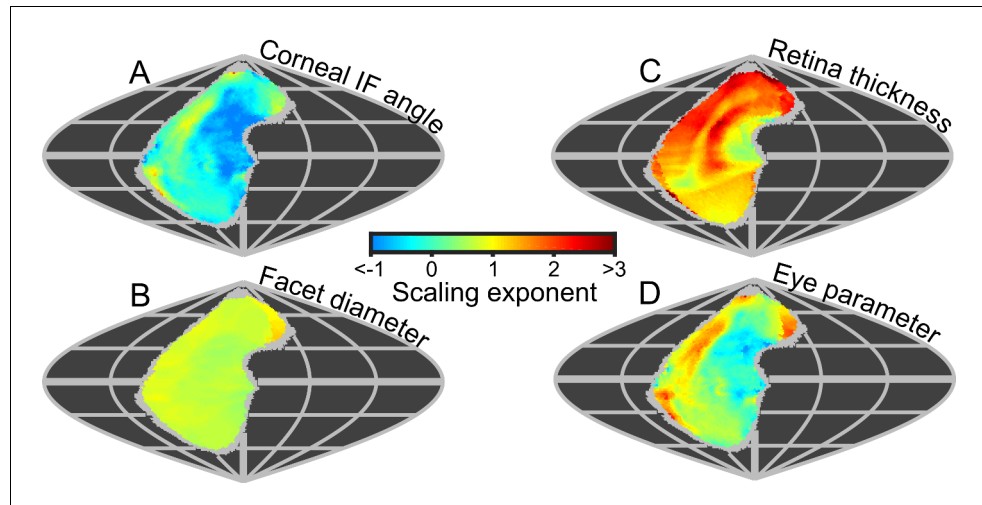

**Figure 6.** Eye metrics and maps of scaling exponents in the visual world. (A–D) Maps of the scaling exponents calculated for bumblebees for the variable specified on each map. The scaling exponents for all variables use the blue-to-red color gradient bar. Positive scaling exponents (red) indicate visual 'improvement' with increasing eye size for the facet diameter (B) and retina thickness (C), whereas conversely, the corneal IF angle (A) improves with eye size given a negative exponent (blue). A positive exponent for eye parameter (D) indicates increasing optical sensitivity, whereas a negative exponent indicates improving optical resolution. We limited the scaling exponent calculations to regions viewed by at least four bees, and the grey fringe in each map indicates the additional regions (viewed by three or fewer individuals) that were not used in the calculations. See *Figure 6—figure supplement 1* for plots of the scaling exponents of lens and CC thickness, and of radius of curvature.
DOI: https://doi.org/10.7554/eLife.40613.014

The following figure supplement is available for figure 6:

**Figure supplement 1.** Scaling and sensitivity.
DOI: https://doi.org/10.7554/eLife.40613.015

indication than the actual topologies themselves of where and how the topology of bumblebee eye properties changes as a function of eye size.

## Eye parameter

Until now, we have described eye properties and their resultant effects on visual capabilities to investigate how bumblebees invest in vision as their eye size increases. The eye parameter ($P$) is an additional metric (calculated as the local IF angle multiplied by the facet diameter) (*Snyder, 1979*) that provides an indication of whether facets are optimized for resolution (where $P$ approaches 0.29 if the resolution of the eye is limited by the diffraction of green light) or sensitivity (where $P$ may be one or higher). Although the average eye parameter increases with eye size (*Figure 3—figure supplement 4A*; we assume IO equals IF angle when calculating $P$), we found that all bumblebees had a similar average eye parameter profile between −30° and 10° (az. - Azimuth) (*Figure 3—figure supplement 4Bii*). Scaling exponent maps showed that the eye parameter of larger bees was slightly reduced in a region similar to that in which their IF angles decreased, but increased in the periphery of the CP (*Figure 6,D*). This indicates that the dorsofrontally improved IF angles correspond to a region of improved optical resolution in larger bees, which strongly suggests that this region of the eye represents an acute zone optimized for high visual resolution (*Land, 1997*).

## Honeybees

Honeybees have slightly larger eye areas and volumes relative to their ITW than bumblebees (*Figure 2A*). When compared on the basis of their EV, the average values for honeybee eyes are generally similar to those for medium-sized *Bombus* (in terms of IF angle (*Figure 3A*), facet diameter (*Figure 3C*), radius of curvature (*Figure 3—figure supplement 1A*), rhabdom length (*Figure 5A*), and eye parameter (*Figure 3—figure supplement 4A*)). However, the distinguishing characteristics

of honeybee eyes are that their values for corneal lens thickness are distinctly lower than those for all bumblebees (*Figure 3—figure supplement 2A*), and that the extent of their binocular overlap is also smaller than that for to medium-sized *Bombus* (*Figure 2C*). The CPs of honeybees were also shifted dorsoposteriorly relative to those of bumblebees (*Figure 4A,B*), while their binocular CP overlap is limited to their dorsal visual field and does not extend frontally (*Figure 4C*). After accounting for the differences in their visual fields, the projected topologies of the honeybees are similar to those of medium-sized bumblebees but with two differences: first, honeybees have an obvious increase in retinal thickness in their lateral visual field (*Figure 5Bii*, −120° to −75° (az. - Azimuth)) and, second, they have relatively small IF angles in their frontal visual field (*Figure 3Bii*, −30° to 30° (az. - Azimuth)), which is the approximate region where larger bumblebees also increase their visual acuity.

## Discussion

In this study, we demonstrated a new microCT-based method for quantifying and mapping the vision capacity of insect eyes. We created 3D models directly from volumetric images of bee's apposition compound eyes to determine how their surfaces are projected into the visual world. This allowed us to determine each eye's CP, across which we projected calculated topologies of the IF angle and of properties influencing optical sensitivity, including lens size and rhabdom thickness. As our method enabled us to project the surface of each eye into the world, we were then able to map the scaling relationships of eye properties and visual capabilities for bumblebees. We used this information to disambiguate how bumblebees of different sizes invest in resolution and sensitivity across their visual field, revealing unexpected patterns in how this investment changes with eye size.

### Comparison of eye analysis techniques

Anatomical studies of insect eyes have typically been based on measurements of 2D representations – either from imaging histological cross-sections or from flattened replicas of the eye surface (*Ribi et al., 1989*). These techniques do allow optical dimensions to be measured from the eye's morphology, but they lose the 3D location of these measurements relative to the remainder of the eye and head. Some studies on compound eyes have also measured 2D approximations of 3D shapes, such as the projected surface area of a compound eye from a given viewpoint (*Spaethe and Chittka, 2003*; *Kapustjanskij et al., 2007*; *Perl and Niven, 2016a*). However, the projected area of a 3D object depends upon on its size, shape, and on the direction of the observer; for example, a hemisphere viewed along its midline appears as a circle with a projected area that is 50% of the true area. Indeed, if the projected area of compound eyes in this study are calculated from orthographic projections along each cornea's principle axis, the true surface area is consistently underestimated by 22%. Unfortunately measurements of projected area are not always clearly distinguished from true surface area in the literature. The method that we developed addresses the limitations of these techniques by maintaining the 3D structure of the eye relative to the head and allowing areas and volumes to be measured directly.

As an alternative to microCT imaging, digital image registration techniques can also be used to reconstruct a volume from serial sections through an insect eye (*Hung and Ibbotson, 2014*). Although a volume reconstructed from sections allows measurements to be made with histological resolution, section distortions and alignment errors may limit the accuracy of calculations that rely on the 3D structure. The radius of curvature of a corneal transect can also be measured from a section (*Schwarz et al., 2011*) or a micrograph of a whole eye (*Bergman and Rutowski, 2016*) and can be combined with a measurement of facet diameter to calculate an approximation of IO angle that is equivalent to our IF angle. When used on butterflies, the micrograph technique closely matches the results of the pseudopupil method. However, measuring the local radius from the eye's profile only provides information along a 2D transect without describing the eye's entire visual field, and calculating the IO angle in this way has similar limitations to our method in the presence of skewed CC (discussed further in the following section and the 'Supplemental methods').

The pseudopupil technique allows IO angle (and facet diameter) to be measured directly from live insects and maintains the topology of these variables with reference to the visual world (*Stavenga, 1979*). Nevertheless, it is rarely applied across the entire FOV (or CP) – the equipment's geometry typically limits the measurement range to ±70° in azimuth and elevation (*Rutowski et al.,*

*2009*) – and cannot measure the dimensions of internal eye structures that influence sensitivity. It is also difficult to measure IO angle directly on compound eyes that lack a pseudopupil and thus have a uniformly black iris pigment (as is the case for many hymenopteran eyes, including those of honeybees and bumblebees). Antidromic illumination is an alternative in such cases that involves the placement of a light source inside an insect's head to create a pseudopupil with light emerging from the eye's facets along the reverse optical path of each receptor. This illumination method has been used to measure IO angles into the frontal visual field of several species (*Kirschfeld, 1973*; *Spaethe and Chittka, 2003*; *Dyer et al., 2016*). However, desert ants and honeybees are the only species in which this method has been used to determine the full FOV, and in the latter case, the IO angle topology (*Seidl, 1982*; *Zollikofer et al., 1995*). The method that we have developed to calculate the CP and IF angle can broadly be applied to any apposition compound eye, regardless of its iris pigmentation, and could even be used on preserved insects with intact corneas. This approach could also be used to estimate the visual properties of superposition eyes (*Land and Nilsson, 2012*), although modifications would be required to account for the larger optical effect of crystalline cones in this eye type. Unique to our approach is the ability to describe vision in world-referenced coordinates, which allows our findings to be compared quantitatively to the results of studies on the vision of other species. In addition, we observed that the topologies of local variables in our analysis were spatially autocorrelated (e.g. *Figure 3B*), and dimensional reduction techniques could be used (with a larger sample size) to determine a subset of parameters that describe the variation in eye shape and visual capabilities observed between bumblebees (*Goodhill et al., 2015*; *Klingenberg, 2016*).

## Comparison to other studies on honeybee and bumblebee eyes

The eye anatomy and visual capabilities of honeybees have previously been studied in detail. As such, they are an important reference species for assessing the validity of the analysis method presented here. In addition, honeybees have no distinct size polymorphism (*Streinzer et al., 2013*), which improves the robustness of comparisons between different studies by minimising differences caused by variation in eye size. As a case in point, the honeybees in our study and those analyzed by *Streinzer et al., 2013* were almost identical in body size (our specimens had ITWs of 2.9 mm and 3.0 mm vs. 2.9 ± 0.0 mm), and had similar eye surface areas (both 2.4 mm$^2$ vs. 2.5 ± 0.1 mm$^2$), facet numbers (5440 and 5484 vs. 5375 ± 143) and maximum facet sizes (25.4 μm and 25.6 μm vs. 25.2 ± 0.3 μm). Thus, we conclude that our method is capable of describing corneal eye properties to within 5% of the values provided by replica-based techniques (with 'worst case' errors of −4.4% for surface area, +4.8% for facet number, and +2.8% for maximum facet diameter). The thicknesses of eye components have previously been described from sections through a honeybee's compound eye (*Greiner et al., 2004*); in the forwards facing areas of our honeybee eyes, we measured similar lens (34 μm and 37 μm vs. 28 μm) and CC thicknesses (46 μm and 50 μm vs. 55 μm). However, our retinal thickness measurements were substantially thinner (219 μm and 223 μm vs. 320 μm), and the value from histology lies above the range that we measured for that parameter. Although our thickness measures are consistent between individuals, the relative difference between our measurement of retinal thickness and that of *Greiner et al. (2004)* is substantially larger than the error we estimated for the corneal eye properties. A likely explanation for this difference is that our definition of thickness is based on the local surface NV from the lens and the distance until it intersects the upper surfaces of the CC, retina, and lamina interfaces (*Figure 1D*), whereas *Greiner et al. (2004)* directly measured the rhabdom length from serial sections. Rhabdom length, rather than retinal thickness, is the relevant dimension for calculating optical sensitivity and, as a rhabdom may not necessarily lie perpendicular to the cornea, its length can evidently exceed our structural thickness measurements (the ratio by which it does so could also vary across the eye). In addition, the mean values and distributions that we calculated for all variables, besides CC thickness (*Figure 3—figure supplement 3*), are closely matched in the two honeybees examined in this study, indicating that our analysis method produces highly repeatable results.

Previous studies on *Apis* have not calculated IF angles, but several have used an antidromically illuminated pseudopupil to measure IO angles directly. Values for the average IO angle in a honeybee's frontal visual field have previously been measured as 1.8° (*Kirschfeld, 1973*) and 2.0° (*Seidl, 1982* from *Giger, 1996*), with the latter recording 1.2° in the acute region. In the frontal and acute regions, the IF angles calculated for our honeybees (frontal: 1.4° and 1.7°; acute: 0.9° and 1.0°) are slightly lower than the reported IO angles, but our measurements exceed those reported

by *Seidl (1982)* in the dorsal and lateral regions (*Supplementary file 1*–Table S2). The primary source of difference between these measurements is that the honeybee's ommatidial viewing axes can be skewed from the corneal NV and this misalignment varies across the eye (*Snyder, 1979*). This skewness is visible in sections through compound eyes (*Baumgärtner, 1928*), and unfortunately is not correctable without segmenting the individual CC for use in optical modeling. As a result of this, our method to calculate IF angles underestimated the smallest honeybee IO angles by approximately 30% (absolute value: −0.3°) but may overestimate other angles by 30% to 60% (+0.8° to +1.3°). Because we calculate the CP from the corneal surface, we likewise underestimate the honeybees' complete FOV as this also depends on CC skewing. An individual eye's FOV is reported by *Seidl and Kaiser (1981)* to be *nearly* hemispherical, whereas the CP of the eyes measured in this study span a quarter (25% and 27%) of the world sphere. The greatest differences between these angular extents appear to occur in the ventral and posterior regions: *Seidl and Kaiser (1981)* reported that the honeybee FOV extends down to −90° in elevation and back to −156° in azimuth (at 0° el.), whereas we find that the CP extends to −60° in elevation (at −60° (az. - Azimuth)) and to −107° in azimuth (at 0° el.). In addition, *Seidl and Kaiser (1981)* found frontal binocular overlap from dorsal to ventral regions, whereas our CP only indicated binocular overlap dorsofrontally. Given that pseudopupil measurements found a larger FOV and IO angles that were often larger than the IF angles from our projection method, the optical axes of ommatidia must generally diverge in honeybee eyes to create a larger FOV at the expense of resolution. Although this demonstrates a limitation in using our method to approximate visual resolution with IF angles, it also highlights the importance of developing additional techniques to determine CC orientation if detailed analyzes and comparisons of visual fields are to be made within and between species.

The visual systems of bumblebees have received less attention than those of honeybees. Nonetheless, several studies have provided data against which we can compare our results. The body sizes of the two medium-sized bumblebees in our study were similar to those of the *B. terrestris* workers analyzed by *Streinzer and Spaethe (2014)* (we measured an ITW of 4.0 mm for both bees, vs. 3.9 ± 0.6 mm, as well as maximum facet sizes of 25.0 μm and 25.2 μm vs. 25.1 ± 1.9 μm). However, the surface areas of our bees' eyes were slightly smaller (2.2 mm$^2$ and 2.4 mm$^2$ vs. 2.8 ± 0.6 mm$^2$) and had fewer facets (4941 and 5505 vs. 5656 ± 475). As both eye and body size vary substantially between bumblebee individuals, we did not estimate their 'worst case' errors as we did for honeybees. Nonetheless, the similarity in the majority of measurements between our study and that of *Streinzer and Spaethe (2014)* suggests good agreement between the different methods used to obtain them and provides further support for the validity of our method. The pseudopupil technique has also been used with antidromic illumination to measure the IO angles from the mediofrontal area of bumblebee eyes as a function their body size (*Spaethe and Chittka, 2003*) for small (ITW: 2.8 mm to 3.0 mm, mean IO angle: 1.5° for six bees) to medium-sized individuals (4.0 mm to 4.2 mm, 1.2° from four bees). By comparison, we found substantially larger IF angles (facing frontally) for our equivalently sized small (3.0 mm, 2.7°) and medium-sized (both 4.0 mm, 2.2° and 2.4°) bees. However, our method shows that the IF angle of bumblebees decreases from their frontal to their lateral visual fields (*Figure 3B*), and that by −45° (az. - Azimuth), the IF angles (small bumblebee 1.5°, medium bumblebee 1.3°) match the IO angle measurements made by *Spaethe and Chittka (2003)*. Eye-referenced locations for IO angle measurements do not provide a clear world-referenced viewing direction, and it is conceivable the measurements of *Spaethe and Chittka (2003)* were taken from eye areas directed somewhat laterally, in which case our results provide similar values. This highlights the importance of using a world-reference for visual studies, even when comparing between individuals of the same species. Regardless of eye size, we found that the minimum IF angles of both bee species were oriented somewhat laterally (*Figure 3Bii*), which also challenges the common assumption that the frontal visual field is always the most relevant for acute insect vision.

## Allometry of *B. terrestris* eye structure

We found a scaling exponent of 0.45 for eye surface area vs. ITW for the *B. terrestris* workers in this study (*Figure 2A*). This is substantially lower than the between species allometry rate found across a range of 11 *Bombus* species that had a similar range of body sizes (0.73) (*Streinzer and Spaethe, 2014*), indicating that the absolute investment in compound eyes varies more between *Bombus* species than between *B. terrestris* individuals. Similar scaling exponents in other *Bombus* species would provide a clear indication that, independently of any factors influencing body size, visual

requirements have influenced the evolution of eye size in different bumblebee species. We also calculated the scaling exponents of facet number and diameter as a function of eye size for the 11 *Bombus* species described by *Streinzer and Spaethe (2014)*. We found that facet number had a substantially higher scaling rate when making comparisons between *Bombus* species (1.39) rather than within *B. terrestris* (0.61). This suggests that the scaling of facet number, and thus resolution, is likely to be a species-specific adaptation. Conversely, the maximum size of facets had a lower scaling rate between species (0.26) than within *B. terrestris* (0.70). A recent study on the allometry of wood ant eyes from different colonies also showed that, despite maintaining similar scaling exponents for total eye size between colonies, two colonies invested in more facets as eye size increased whereas another invested in larger facets (*Perl and Niven, 2016a*). Evidently, varying the parameters of eye allometry allows for substantial fine-tuning of the visual performance of individuals, both between and within species. The allometry of visual performance in bumblebees may influence which individuals and species can most effectively forage for specific floral resources (*Dafni et al., 1997*).

## Allometry of *B. terrestris* visual capabilities

Within a species, variation between individuals' visual capabilities could impact on their relative foraging ability. Behavioural studies investigating the influence of size on visual performance have shown that larger bumblebees (i) do indeed have more acute vision when trained to discriminate visual targets (*Spaethe and Chittka, 2003*), (ii) are able to fly at lower light intensities (*Kapustjanskij et al., 2007*), and (iii) are more efficient foragers (*Spaethe and Weidenmüller, 2002*). *Spaethe and Chittka (2003)* also found that an increase in body size of 34% halved the minimum angular object size that a bee could identify, yet they noted that the allometric improvement in IO angle that they measured could not directly predict the improvement in behavioural visual acuity. We applied the linear regression equation calculated by *Spaethe and Chittka (2003)* (*Angle* (°) = 17.6–3.1 × *ITW* (mm)) to predict the minimum detectable object size for the small and medium bees in our study (ITWs 3.0 mm and 4.0 mm), giving visual angles of 8.4° and 5.2°, a 38% decrease. While these angles are substantially larger than the IF angles calculated using our method, the greatest relative improvement in IF angle (at any matching direction in the common CP) between our small and medium-sized bees is 38%, surprisingly close to the relative 34% improvement predicted by the behavioural study. This local acuity improvement is directed frontally (−9° el., −15° (az. - Azimuth)), in a region of the visual field that is well positioned for target discrimination and that has been shown to be important for measuring optic flow for flight control (*Baird et al., 2010*; *Linander et al., 2015*). Our findings suggest that it may in fact be possible to predict the relative differences in behavioural measures of visual acuity (on insects of different sizes) by measuring the allometry of visual resolution at the relevant location in the visual field.

Interestingly, bigger eyes do not always provide increased visual performance – bumblebees with larger eyes do not exhibit differences when discriminating between different periodic patterns (*Chakravarthi et al., 2016*), although the range of body sizes (3.2– mm to 4.3 mm) was narrower than that of the bees analyzed in the present study. Given that the IO angle on the mediofrontal eye area of medium-sized bumblebees is 1.2° (*Spaethe and Chittka, 2003*), relatively poor resolution limits have been measured for target detection (2.3°) (*Dyer et al., 2008*; *Wertlen et al., 2008*) and pattern discrimination (4.8°/cycle) (*Chakravarthi et al., 2016*). The limits obtained from these behavioural experiments are approximately twice those that would be expected on the basis of the sampling frequency (*Snyder, 1979*). Where the cut-off frequency of the optics is lower than the sampling frequency of the ommatidial array, the optics of the compound eye lenses may also limit visual acuity through oversampling, an additional limiting factor that we have not considered here (*Snyder, 1979*). Our analysis shows that lens thickness does vary across all bees' eyes (*Figure 3—figure supplement 2B*), which is likely to result in local differences in focal length and may lead to topological variation in acceptance angle and, thus, the optical cut-off frequency. An approximately 25% increase in acceptance angle has indeed been found between frontally and laterally facing ommatidia in honeybees (*Rigosi et al., 2017*). Two additional considerations when determining visual acuity from anatomical measurements are that the acceptance angle of many insects varies between states of light and dark adaptation (*Warrant and McIntyre, 1993*), and that both object illumination and contrast also influence an eye's effective acuity (*Snyder et al., 1977*; *Warrant, 1999*).

## Allometry of *B. terrestris* visual fields

In this study, we found that the CP of bumblebee eyes increased with eye size (*Figure 2C*). This was a consequence not simply of having a larger eye, but also of a change in eye shape such that the surface was projected onto a larger angular area. We also identified an area of binocular overlap not previously reported in bumblebees. The extent of this corneal binocular overlap, directed both frontally and dorsofrontally (*Figure 4C*), increased rapidly with body size. Bumblebee workers have been found to approach artificial (*Reber et al., 2016*) and natural flowers (*Orth and Waddington, 1997*) from below, which would place the visual target dorsofrontally, in a region where we also found that IF angle decreases with eye size (*Figure 6A*). Hence, larger bumblebees would view the flowers they approach with a more acute and larger binocular visual field, which would potentially improve their visual discrimination or landing control relative to that of smaller bees. The potential benefits of this binocular overlap would be an interesting topic for further behavioural investigations.

Surprisingly, the scaling exponent (as a function of body size) found here for bumblebees' CP is nearly identical to that found from the optically measured FOV of differently sized butterfly species (*Rutowski et al., 2009*). This is the case for both the visual field of a single eye (we found 0.17 vs. 0.14) and the binocular regions (0.79 vs. 0.82). By contrast, the FOV of desert ants remains similar over a nearly two-fold increase in head size (*Zollikofer et al., 1995*). Given the common scaling rates shared by the *B. terrestris* worker's CP and that of butterflies, we hypothesise that increasing the visual field of each eye (and the binocular overlap between eyes) at the identified rates is a general strategy for compound eye enlargement among different groups of flying insects. Although visual field extent has rarely been considered during previous studies of insect vision, increasing FOV size has been shown to improve the performance of visually guided behaviours such as navigation (*Wystrach et al., 2016*) and visual motion detection (*Borst and Egelhaaf, 1989*), and is likely to improve the ability of larger bumblebees to perform these visually guided behaviours.

## Allometry of *B. terrestris* visual topology

Local variation in the scaling rate of eye properties will cause eye-dependent variation in the topology of visual capabilities. The region with the lowest IF angle scaling exponent (leading to improved visual resolution) is directed dorsofrontally (*Figure 6A*), while a positive but relatively similar scaling exponent for facet size occurs across the visual field (*Figure 6B*). By contrast, the scaling rate of IO angles vs. body size of Orange Sulphur butterflies was greatest in the ventral, ventrofrontal, and dorsal eye areas, while their facet diameters were also found to increase at a uniform rate across the areas measured (*Merry et al., 2006*). Again, the results from ants are qualitatively different from those for bees and butterflies: the scaling exponent of facet diameter varied between the eye areas of wood ants, being highest in the dorsal and frontal areas (*Perl and Niven, 2016b*), whereas a study on desert ants found that IO angle scaled similarly between lateral and dorsal eye areas (*Zollikofer et al., 1995*). Pseudopupil measurements along a vertical transect of the eyes of damselfly species found that the maximum diameters and minimum IO angles were influenced by both eye size and habitat (*Scales and Butler, 2016*). Although the scaling exponents along the eye transects were not measured, damselflies living in dim, cluttered habitats appeared (independently of eye size) to have more prominent eye specializations than those living in open habitats.

To our knowledge, this is the first study to investigate the topology of retinal thickness in insects (*Figure 5*),and it is evident from our analysis that retinal thickness varies substantially across all bee eyes. If differences in retinal thickness are translated into equivalent differences in rhabdom length, this would influence optical sensitivity across each eye by 20–50% for bumblebees and by 53% for honeybees (based on the difference between the minimum and maximum retinal thickenss of each eye and the influence of rhabdom length on sensitivity). Retinal thickness is typically highest ventrally and posteriorly (*Figure 5B*), where higher retinal sensitivity may compensate for the reduced effective aperture that results from the skewed CC in these regions (*Stavenga, 1979*). Retinal thickness has a positive scaling exponent across the majority of the visual field (*Figure 6C*), which would improve the optical sensitivity of larger bees. Unexpectedly, we identified that retinal thickness increases at a greater rate in the dorsal hemisphere and would provide larger bumblebees with relatively increased dorsal sensitivity that may, for instance, assist a bee's ability to visually discriminate downwards facing flowers that are not directly illuminated by sunlight (*Makino and Thomson, 2012*; *Foster et al., 2014*). Our results demonstrate that, in addition to facet size, retinal dimensions offer

substantial scope for insects to fine-tune optical sensitivity across their visual fields, a point that appears to have been overlooked by previous studies.

The visual topologies and scaling exponents that we measured for bumblebees partially support our initial hypothesis that the increased area of a larger eye would be invested primarily in improving the capabilities of a small visual region. The improved visual resolution of larger bees is primarily directed dorsofrontally, but the scaling of facet diameter and retinal thickness would lead to increased optical sensitivity across their entire field of view. As a result, we now hypothesise that, for a given insect group, specific regions in the visual field may have certain 'ideal' requirements for resolution and/or sensitivity that are based on the visual information available in their specific habitat and their behavioural ecology. Once the size of an eye allows such a threshold to be reached, additional area could then be more broadly invested in improving the visual capabilities across the visual field. This revised hypothesis incorporates our findings that the differential allometry of *Bombus* eye properties allows their visual capabilities to be improved both locally or globally across their growing visual field.

## Conclusion

Analysing the 3D structure of insect eyes to determine a holistic description of their visual capabilities provides insight into how the morphology of eyes has evolved to sample visual information from the world. We find that the differential scaling of the morphology between eye areas allows bigger bumblebees to invest the increased resources of a larger eye in improved sensitivity across an enlarged visual field. Yet, studying the allometry of bumblebees' entire visual topology also indicated specific regions that have a high investment rate, such as the dorsofrontal region of both enlarging binocularity and increasing resolution, or the high rate of thickening in the dorsally facing retina. Important visual information is presumably viewed by bumblebees in these regions of their visual fields, indicating a promising avenue for further behavioural experiments, such as the use of virtual reality to manipulate the visual cues at specific regions in an insect's FOV (*Stowers et al., 2017*) and observational studies to identify what bumblebees view in those regions when flying through natural environments (*Stürzl et al., 2015*). The differential allometry between eye areas undoubtedly endows insects that have larger eyes with improved vision because they have a better capacity to match their visual capabilities to the requirements of both their environment and their behaviour.

## Materials and methods

### Study animals

Bumblebees (*Bombus terrestris*) spanning the typical range of body sizes (categorized here as small, medium, and large) were collected from a commercial hive (Koppert, UK). Honeybees (*Apis mellifera*) were collected from hives maintained at the Department of Biology, Lund University, Sweden. Several workers of each species and size category were collected and anesthetized with carbon dioxide gas before being dissected.

### Sample preparation

Samples were preserved by dissecting the left compound eye (to preserve this alone), or by removing the front, bottom, and rear of the head capsule (to preserve the whole head). They were then fixated, stained, and embedded in epoxy resin. See the 'Supplemental methods' for further information about the preparation procedure. We also fixated several completely intact heads of each bee species, which we then dehydrated in ethanol and critical point dried. The inter-tegula width (ITW) of each sample was measured with digital callipers to provide a measure of body size (*Figure 1D*) (*Cane, 1987*).

### X-ray microtomography

Tomographic imaging of samples was conducted at the Diamond-Manchester Imaging Branchline I13-2 (*Rau et al., 2011*; *Pešić et al., 2013*) at the Diamond Light Source, UK. See the 'Supplemental methods' for further information about the imaging parameters. Dissected and dried heads were

imaged using ×2.5 total magnification (2.6 µm effective pixel size, *Figure 1B*), whereas isolated eyes were imaged with ×4 total magnification (1.6 µm effective pixel size, *Figure 1D*).

## Volumetric analysis

We examined the imaged volumes to choose, for further analysis, the two best-preserved compound eye samples from each species and size category (six bumblebees and two honeybees in total) and the best-preserved head capsule from each species. Amira (FEI) was used to analyze these volumes in three ways: i) by manually labeling the structures of a compound eye (*Figure 1A*), ii) by aligning the labeled compound eyes of a given species onto the scan of a full head (*Figure 1B*), and iii) by measuring the facet dimensions on a compound eye (*Figure 1C*). Additional details of the procedure used to process volumes in Amira are provided in the 'Supplemental methods'.

## Computational analysis

We developed Matlab scripts to use data from the volumetric analysis performed in Amira (labeled volumes, facet measurements and transforms) to compute the eye properties (eye surface area, eye volume, facet number, *facet diameter*, *radius of curvature*, and *thicknesses* (for the lenses, CC, and retina)), visual capabilities (individual CP, binocular CP, complete CP, *IF angle*, and *optical sensitivity*) and a metric (*eye parameter*). Note that the calculations of optical sensitivity and eye parameter assume that the IO angle equals the computed IF angle (which is not met across the entire eye (*Figure 1D*)), limiting the accuracy with which we can report these parameters. The italicized variables were determined locally, that is, they were calculated at sampling points that were equally spaced at 25 µm intervals across each bee's corneal surface. The corneal normal vector (NV) of each sampling point was used as an indication of the viewing direction of that part of the eye in space (*Figure 1D*). We determined which viewing directions occurred inside the CP of each bee (*Figure 1E*), before interpolating each locally calculated variable from the sampling points onto the world. This allowed locally calculated variables to be represented in both eye- and world-centric coordinates, which are reported using plots of: facet-wise mean values and distributions, projections of CP limits (and topologies) onto the world, and profiles across both azimuth or elevation representing the average of a projected parameter (or the integral of CP) across 10° bands of visual space. Additional details about this computational analysis procedure and a discussion on its limitations are provided in the 'Supplemental methods'. The MATLAB scripts for calculating and plotting eye properties are available for download (*Taylor, 2018*; copy archived at https://github.com/elifesciences-publications/compound-eye-plotting-elife).

## Allometry

The allometry of values calculated from bumblebee eyes was described by fitting the parameters $b$ and $\alpha$ in the power function $Y = bx^{\alpha}$ (*Huxley and Teissier, 1936*), after a logarithmic transformation of the size indicator ($x$) and the dependent variable ($Y$). See the 'Supplemental methods' for further information about the allometry procedure. Allometry functions were calculated for a given variable (if the variable was calculated locally then the facet-wise mean value was used) measured for all bumblebees (*Supplementary file 1*–Table S1). As the topology of most parameters varied across the eye, we also calculated local allometric functions for variables on the basis of their projection into the world and represent these as spatial maps of the scaling exponent of each variable (*Figure 6* and *Figure 6—figure supplement 1*).

## Acknowledgments

We would like to thank Carina Rasmussen, Eva Landgren, and Ola Gustafsson for facilitating sample preparation and providing access to the Microscopy Facility at the Department of Biology, Lund University. We are also grateful to Karin Odlén, Julia Källberg, Per Alftrén, and particularly Viktor Håkansson, for assistance preparing samples and analysing data, and also to Viktor for diligently managing tomographic reconstructions during our beamtime. Our imaging was performed at Diamond Light Source (proposals 13848 and 16052), where we were pleased to receive assistance from Qiang Tao, David Wilby, Rajmund Mokso, and Kazimir Wanelik. Gavin Taylor is thankful to have received a stipend from Carl Tryggers Stiftelse (CTS15:38) and an endowment from the Royal Physiographic Society of Lund. Emily Baird acknowledges financial support from the Air Force Office of

Scientific Research (FA8655-12-1-2136), the Swedish Research Council (2014–4762), and the Lund University Natural Sciences Faculty. Pierre Tichit received funding from the Interreg Project LU-011, and Marie Schmidt received funding from the Erasmus + program.

## Additional information

### Funding

| Funder | Grant reference number | Author |
|---|---|---|
| Air Force Office of Scientific Research | FA8655-12-1-2136 | Emily Baird |
| Vetenskapsrådet | 2014-4762 | Emily Baird |
| Carl Tryggers Stiftelse för Vetenskaplig Forskning | CTS15:38 | Gavin J Taylor |
| Kungliga Fysiografiska Sällskapet i Lund | | Gavin J Taylor |
| Lund University Natural Sciences Faculty | | Emily Baird |
| Interreg Europe | LU-011 | Pierre Tichit |
| Erasmus+ | | Marie D Schmidt |

The funders had no role in study design, data collection and interpretation, or the decision to submit the work for publication.

### Author contributions

Gavin J Taylor, Conceptualization, Resources, Data curation, Software, Formal analysis, Supervision, Funding acquisition, Investigation, Visualization, Methodology, Writing—original draft, Project administration, Writing—review and editing; Pierre Tichit, Data curation, Formal analysis, Investigation, Writing—review and editing; Marie D Schmidt, Conceptualization, Data curation, Software, Formal analysis, Investigation, Methodology, Writing—review and editing; Andrew J Bodey, Resources, Formal analysis, Investigation, Methodology, Writing—review and editing; Christoph Rau, Resources, Investigation, Methodology; Emily Baird, Conceptualization, Resources, Supervision, Funding acquisition, Investigation, Writing—original draft, Project administration, Writing—review and editing

### Author ORCIDs

Gavin J Taylor http://orcid.org/0000-0003-4787-9844
Pierre Tichit http://orcid.org/0000-0003-0310-6073
Emily Baird http://orcid.org/0000-0003-3625-3897

### Decision letter and Author response

Decision letter https://doi.org/10.7554/eLife.40613.062
Author response https://doi.org/10.7554/eLife.40613.063

## Additional files

### Supplementary files

• Supplementary file 1. Supplemental tables (Tables S1 and S2) for the manuscript.
DOI: https://doi.org/10.7554/eLife.40613.016
• Transparent reporting form
DOI: https://doi.org/10.7554/eLife.40613.017

### Data availability

The original data calculated from eye volumes (and used in the plots of this paper) and a reformatted version have both been uploaded to Dryad (https://dx.doi.org/10.5061/dryad.23rj4pm). The

original image data of the raw and labelled volumes for each of the imaged compound eyes (and heads) are available via MorphoSource.

The following datasets were generated:

| Author(s) | Year | Dataset title | Dataset URL | Database and Identifier |
|---|---|---|---|---|
| Taylor GJ, Tichit P, Schmidt MD, Bodey AJ, Rau C, Baird E | 2018 | Data from: Bumblebee visual allometry results in locally improved resolution and globally improved sensitivity | https://dx.doi.org/10.5061/dryad.23rj4pm | Dryad Digital Repository, 10.5061/dryad.23rj4pm |
| Taylor GJ, Tichit P, Schmidt MD, Bodey AJ, Rau C, Baird E | 2018 | Left compound eye, raw volume, ITW=2.9mm (Speciman: LU:3_14:AM_F_5, Apis mellifera) | https://doi.org/10.17602/M2/M65646 | MorphoSource, 10.17602/M2/M65646 |
| Taylor GJ, Tichit P, Schmidt MD, Bodey AJ, Rau C, Baird E | 2018 | Left compound eye, labelled volume ITW=2.9mm (Specimen: LU:3_14:AM_F_5, Apis mellifera) | https://doi.org/10.17602/M2/M65317 | MorphoSource, 10.17602/M2/M65317 |
| Taylor GJ, Tichit P, Schmidt MD, Bodey AJ, Rau C, Baird E | 2018 | Left compound eye, raw volume, ITW=2.95mm (specimen: LU:3_14:AM_F_8, Apis mellifera) | https://doi.org/10.17602/M2/M65648 | MorphoSource, 10.17602/M2/M65648 |
| Taylor GJ, Tichit P, Schmidt MD, Bodey AJ, Rau C, Baird E | 2018 | Left compound eye, labelled volume, ITW=2.95mm (Specimen: LU:3_14:AM_F_8, Apis mellifera) | https://doi.org/10.17602/M2/M65318 | MorphoSource, 10.17602/M2/M65318 |
| Taylor GJ, Tichit P, Schmidt MD, Bodey AJ, Rau C, Baird E | 2018 | Left compound eye, raw volume, ITW=1.97mm (Specimen: LU:4_16_:BT_F_CE_13, Bombus terrestris) | https://doi.org/10.17602/M2/M65655 | MorphoSource, 10.17602/M2/M65655 |
| Taylor GJ, Tichit P, Schmidt MD, Bodey AJ, Rau C, Baird E | 2018 | Left compound eye, labelled volume, ITW=1.97mm (LU:4_16_:BT_F_CE_13, Bombus terrestris) | https://doi.org/10.17602/M2/M65323 | MorphoSource, 10.17602/M2/M65323 |
| Taylor GJ, Tichit P, Schmidt MD, Bodey AJ, Rau C, Baird E | 2018 | Left compound eye, raw volume, ITW=2.97mm (Specimen: LU:4_16_:BT_F_CE_14, Bombus terrestris) | https://doi.org/10.17602/M2/M65656 | MorphoSource, 10.17602/M2/M65656 |
| Taylor GJ, Tichit P, Schmidt MD, Bodey AJ, Rau C, Baird E | 2018 | Left compound eye, labelled volume, ITW=2.97mm (Specimen: LU:4_16_:BT_F_CE_14, Bombus terrestris) | https://doi.org/10.17602/M2/M65324 | MorphoSource, 10.17602/M2/M65324 |
| Taylor GJ, Tichit P, Schmidt MD, Bodey AJ, Rau C, Baird E | 2018 | Left compound eye, raw volume, ITW=4mm (Specimen: LU:4_16_:BT_F_CE_10, Bombus terrestris) | https://doi.org/10.17602/M2/M65653 | MorphoSource, 10.17602/M2/M65653 |
| Taylor GJ, Tichit P, Schmidt MD, Bodey AJ, Rau C, Baird E | 2018 | Left compound eye, labelled volume, ITW=4mm (Specimen: LU:4_16_:BT_F_CE_10, Bombus terrestris) | https://doi.org/10.17602/M2/M65322 | MorphoSource, 10.17602/M2/M65322 |
| Taylor GJ, Tichit P, Schmidt MD, Bodey AJ, Rau C, Baird E | 2018 | Left compound eye, raw volume, ITW=4.02mm (Specimen: LU:4_16:BT_F_CE_11, Bombus terrestris) | https://doi.org/10.17602/M2/M65649 | MorphoSource, 10.17602/M2/M65649 |
| Taylor GJ, Tichit P, Schmidt MD, Bodey AJ, Rau C, Baird E | 2018 | Left compound eye, labelled volume, ITW=4.02mm (Specimen: LU:4_16:BT_F_CE_11, Bombus terrestris) | https://doi.org/10.17602/M2/M65319 | MorphoSource, 10.17602/M2/M65319 |
| Taylor GJ, Tichit P, Schmidt MD, Bodey AJ | 2018 | Left compound eye, raw volume, ITW=5.42mm (Specimen: LU:4_16_:BT_F_CE_3, Bombus terrestris) | https://doi.org/10.17602/M2/M65652 | MorphoSource, 10.17602/M2/M65652 |
| Taylor GJ, Tichit P, Schmidt MD, Bodey AJ, Rau C, Baird | 2018 | Left compound eye, labelled volume, ITW=5.42mm (Specimen: LU:4_16_:BT_F_CE_3, Bombus | https://doi.org/10.17602/M2/M65321 | MorphoSource, 10.17602/M2/M65321 |

| Authors | Year | Description | DOI | Database |
| --- | --- | --- | --- | --- |
| E | | terrestris) | | |
| Taylor GJ, Tichit P, Schmidt MD, Bodey AJ, Rau C, Baird E | 2018 | Left compound eye, raw volume, ITW=5.47mm (Specimen: LU:4_16_:BT_F_CE_1, Bombus terrestris) | https://doi.org/10.17602/M2/M65651 | MorphoSource, 10.17602/M2/M65651 |
| Taylor GJ, Tichit P, Schmidt MD | 2018 | Left compound eye, labelled volume, ITW=5.47mm (Specimen: LU:4_16_:BT_F_CE_1, Bombus terrestris) | https://doi.org/10.17602/M2/M65320 | MorphoSource, 10.17602/M2/M65320 |
| Taylor GJ, Tichit P, Schmidt MD, Bodey AJ, Rau C, Baird E | 2018 | Head, raw volume (Specimen: LU:3_14_:AM_F_3, Apis mellifera) | https://doi.org/10.17602/M2/M65659 | MorphoSource, 10.17602/M2/M65659 |
| Taylor GJ, Tichit P, Schmidt MD, Bodey AJ, Rau C, Baird E | 2018 | Head, labelled volume (Specimen: LU:3_14_:AM_F_3, Apis mellifera) | https://doi.org/10.17602/M2/M65326 | MorphoSource, 10.17602/M2/M65326 |
| Taylor GJ, Tichit P, Schmidt MD, Bodey AJ, Rau C, Baird E | 2018 | Head, raw volume (Specimen: LU:4_14:BT_F_5, Bombus terrestris) | https://doi.org/10.17602/M2/M65657 | MorphoSource, 10.17602/M2/M65657 |
| Taylor GJ, Tichit P, Schmidt MD, Bodey AJ, Rau C, Baird E | 2018 | Head, labelled volume (Specimen: LU:4_14:BT_F_5, Bombus terrestris) | https://doi.org/10.17602/M2/M65327 | MorphoSource, 10.17602/M2/M65327 |

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

## Appendix 1

DOI: https://doi.org/10.7554/eLife.40613.018

### Supplemental methods

### List of abbreviations

az. – Azimuth
BP – Border point
CC – Crystalline cone
CLP – Corneal linkage point (of a facet dimension measurement)
CP – Corneal projection
D – Facet diameter
el. – Elevation
EV – Eye volume
FOV – Field of view
IDW – Inverse distance weighted
ITW – Inter-tegula width
IO – Inter-ommatidial
IF – Inter-facet
NV – Normal vector
R – Radius of curvature
SP – Sampling point
WP – World point
$\Delta\Phi$ – IF angle
$\sigma$ – Facet axis density

### Sample preparation

Dissected samples were immediately fixed in 3% paraformaldehyde, 2% glutaraldehyde, and 2% glucose in phosphate buffer (pH ~7.3, 0.2M) for 1 to 3 hr, and then washed in buffer before being immersed in 2% $OsO_4$ for 1 hr to enhance the x-ray absorption contrast of soft tissues (*Ribi et al., 2008*). After washing in buffer again, the samples were dehydrated with a graded alcohol series, and acetone was used to transition the samples to epoxy resin (Agar 100), which was cured in an oven at 60°C for ~48 hr. The samples in wet epoxy were placed on Perspex sticks, and after drying, the external resin was peeled to expose the cuticle of the sample (*Taylor et al., 2016*).

### X-ray microtomography

At the beamline, an undulator (gap set to 5 mm) was used to produce a polychromatic (8 to 35 keV) 'pink' beam of partially coherent, near-parallel, x-rays with lower energies suppressed using metal filters. We collected projection images of each sample from 4001 equally spaced angles over 180° of continuous rotation with a scintillator coupled pco.edge 5.5 (PCO AG) detector. A propagation (sample to scintillator) distance of 100 mm was used to give a moderate level of inline phase contrast. Projection images were flat- and dark-field corrected prior to reconstruction into 3D volumes using a filtered back projection algorithm that incorporated ring artefact suppression (*Basham et al., 2015*; *Titarenko, 2016*).

### Volumetric analysis

The reconstructed 32-bit volumes were initially cropped spatially (around the scanned feature), and in their intensity range using the program Drishti Paint (*Limaye, 2012*), and re-saved as 8-bit files. Volumes for labeling and registration were resampled to 5 µm voxel size using a

Mitchel filter (in Amira), but resampling was not used for facet measurement as this could prevent the facets borders from being visualized. Labeling was performed by using the 'brush' tool in the Amira Segmentation Editor to delineate manually the borders of the three gross anatomical structures in a compound eye (lenses, crystalline cones (CC), and retina; *Figure 1A*) from individual slices through the grayscale volume. Features were labeled at 5 to 10 slices intervals, after which an interpolation tool was applied between the labeled slices. High contrast existed between the exterior of the lenses and the surrounding air and this border could be automatically delineated using a threshold tool. The 3D label volumes were smoothed after the manual segmentation and interpolation steps. The interfaces between the lenses-to-exterior, the lenses-to-CC, and the CC-to-retina were selectively labeled by using morphological operations between each pair of labels. The interface between the retina and lamina was identified by selectively rendering the volume immediately proximal to the labeled retina and also by viewing a surface generated from the retina. The surface path selector was then used to trace a closed outline of the lamina interface on the vertices of the retinal surface; the path could then be filled to select all enclosed vertices, which were converted to labeled voxels in the main label volume. The label volume was saved as a series of images for later analysis in Matlab.

The alignment of labeled compound eyes onto a head of the same species was performed manually in the Amira Project Viewer (*Figure 2A*). A single head was used to align all compound eye replicates of a given species, and its grayscale volume was initially loaded into Amira and oriented such that the front of the head faced forwards and the tops of the eyes were parallel to the global coordinates of Amira. The grayscale volume of each labeled left compound eye was loaded, and then either an isosurface or a volume rendering was used to visualize both the head and compound eye (depending on which visualization method provided the clearest representation of its structure). Each eye was then positioned and oriented such that it overlaid the matching compound eye on the head; this also required isometrically re-scaling the head for bumblebee eyes, as a single head from a medium-sized bee was a used as a reference to register the six compound eyes taken from differentially sized individuals. This re-scaling implicitly assumed that the relative position of the compound eyes on the head capsule did not vary between bees with different body sizes. Once aligned, the digital volume was resampled into the world coordinate frame, then flipped about its x-axis (equivalent to mirroring about the sagittal plane of the head), and the mirrored eye was then positioned and aligned onto the head's right eye. The affine transformation matrices for the head, and both the original and mirrored eyes, were then recorded from the Amira console.

Facet dimensions were measured on compound eyes that were visualized using either an isosurface or volume rendering. Although the borders of individual facets were rarely visible in individual slices through the volume, these surface-visualization techniques typically showed the fine borders between the facets. However, not all facets were visible on a given eye because of the unavoidable presence of unpeeled glue or dust on the eye, which prevented facet visualization in some areas. Points were selected at the centre of the furthest sides of neighboring ommatidial lenses to measure the size of a given facet. Points were selected in pairs, such that each represented a line passing across the middle of the central facet to link the opposing sides of two neighboring ommatidial lenses (*Figure 1C*). Twenty or more measurements were taken on any individual eye (depending on its size) and we aimed to distribute these evenly across the eye. We avoided including highly irregular facets (i.e. where one hexagon side length was much shorter than the others) in the analysis. The 3D coordinates of each selected point were saved in spreadsheets for import into Matlab.

## Calculation of global parameters

Data saved from Amira (labeled volumes, transforms, and facet measurements) were imported into Matlab with the metadata for each bee (species, body size, etc.). After loading the labels, the eye volume (EV) was calculated from the total number of labeled voxels (including the lens, CC, retina and lamina contact labels, and their interfaces). We found that the most

accurate method for measuring the total eye surface area was to take half of the surface area of an isosurface fit around the corneal surface voxels.

## Transformation and alignment

The coordinates of the eye were transformed from volume coordinates into the head coordinate frame using the recorded transform, and mirrored to produce the right eye (which was also transformed appropriately). The three principal axes were then calculated from the corneal surface coordinates of both eyes, and both eyes were rotated such that the corneal surfaces were symmetric in the sagittal (roll) and transverse (yaw) planes. The vertical axis of the eyes (representing the coronal (pitch) plane) were then aligned to the frontal world horizontal axis.

## Facet shape determination

The six points measured for every facet were transformed into the same coordinate frame as the head, and the centre of the points measured for each facet was linked to the closest point on the corneal surface, which was termed the corneal linkage point (CLP). Locally, the corneal surface was treated as a flat (2D) plane on which the hexagonal facets were arranged. However, the measurements points of each facet were in 3D coordinates and oriented in different directions, so the world's azimuth and elevation directions were chosen as a 2D coordinate system onto which all facets were flattened. To facilitate this, the normal vector (NV) was calculated at the CLP for each facet and the measurement points were rotated such that this vector pointed forwards in world coordinates. Each pair of measured points represented a line between the opposing sides of three adjacent facets, and we reduced the length of each line by 2/3 such that its ends represented the centre points of the outer two facets in a trio (assuming that the two facet centres were symmetrically placed around the central facet). Thus, we determined seven points for each measured facet that represented its own centre and the centre of each neighboring facet. We used this information to compute a Voronoi diagram in which the central cell indicated the corner points of the measured facet (*Figure 1C*). Although the corner points of each facet could have been measured directly, our method effectively averages the shape of the seven facets covered by the initial six measurement points and is conceptually similar to the practice of averaging facet diameter (*D*) over a row of five facets (*Streinzer et al., 2013*; *Streinzer and Spaethe, 2014*; *Streinzer et al., 2016*). The area of the resulting hexagonal facet was then calculated.

## Sampling variables on the eye

Sampling points (SP) were selected across the 3D corneal surface of each eye at approximately equally spaced (25 µm) intervals and formed the basis for eye-centric calculations. At each SP, we calculated the NV of the corneal surface and the coordinates of its intersection on a distant sphere that represented the visual world (*Figure 1D*). The direction of each NV was then reversed to calculate the thicknesses beneath a given SP. We calculated the distance from the SP on the cornea to the vector's intersection point with the lens-to-CC (L-CC) interface (local lens thickness), and then between the L-CC intersection and the subsequent intersection with the CC-to-retina (CC-R) interface (local CC thickness). If the vector subsequently intersected the retina-to-lamina (R-La) interface, we calculated the distance to that intersection from the CC-R intersection (local retina thickness). If the vector did not intersect the R-La interface, we instead found the closest voxel on the R-La interface to the CC-R intersection and calculated the distance between these two points as the retina thickness. We also calculated the NVs and sphere intersections from all voxels around rim of the corneal surface (where the cornea bordered the surrounding cuticle), although we did not calculate thicknesses for these border points (BP).

## Normal vector calculations from voxelized surfaces

We determined the corneal NV at each SP by selecting all corneal surface voxels within a local radius around the SP and fitting a 2$^{nd}$ order polynomial surface to the selected voxels, and then calculated the NV from the derivative of the surface at the SP (*Taylor et al., 2016*). This method of determining a surface's NV was implemented in a similar manner to the 'coordinate transform' method of parameterizing discretized surfaces used in computer vision applications (*Stokely and Wu, 1992*). The choice of local radius determined the number of points included in the surface fitting, which influenced the NV and caused noticeable variation in the calculated inter-facet (IF) angle, as this was the result of finding the angle between multiple NVs (*Figure 1D*). An iterative procedure that allowed us to choose an appropriate local radius for NV calculations on each eye is described in a following section. We also noticed that particularly flat corneal surfaces lead to inaccurate normal fitting, as information on the surface's shape was lost when discretising the eye into voxels. To rectify this problem, we tested whether the outermost voxels in each selection varied in height with respect to the central voxels; if no variation was present (i.e. a flat plane of voxels had been selected), then we enlarged the local radius until height variation was present across the selected points (see the final section for further discussion about this).

## Interpolation of hexagons between sampling points

The hexagon parameters at each SP were interpolated from the sparser set of CLPs using inverse distance weighted (IDW) interpolation (with p=3), where distances between points were based on the geodesic path lengths over the corneal surface (*Shepard, 1968*). We devised a shape interpolation method based on the transformation vectors between the closest corner points of two centred hexagons. The corner points of the hexagon CLP2 could then be described from the original hexagon's corner points (CLP1) plus the transform (T), as CLP2 = CLP1+T. As the facet shape determined for each CLP was initially rotated to face forwards, all corner points lay on the same plane (allowing 2D transformations to be used) and each facet's alignment with respect to the world's azimuth and elevation was preserved. These hexagon transformations were calculated between every pairwise combination of CLPs, and used to interpolate the shape of a hexagon at a given SP in the following manner:

- For each SP, only nearby CLP pairs (where both were within 100 µm distance) were used in the interpolation.
- For every pair of CLPs, the ratio of the distance between them and the SP was used to determine the magnitude of the transformation applied to CLP1. That is, the predicted hexagon (H) was, $H = CLP1 + T \times G_{SP \to CLP1}/(G_{SP \to CLP1} + G_{SP \to CLP2})$, where G denotes the geodesic path length between the two specified points. Hence, if the SP was closer to CLP1, H would be similar to CLP1, or vice versa if the SP was closer to CLP2.
- Predicting a hexagon from N nearby CLP pairs produced 6N corner points for a given SP. These were divided into six groups using the K-means clustering algorithm.
- The (weighted) average coordinate of each group then provided the six corner points of the interpolated hexagon at a given SP. The weight for each predicted hexagon was the inverse of the average distance for both CLPs to the SP.
- The area of the interpolated hexagon was calculated, as was the distance and angle to the six adjacent facet centres based on treating the interpolated hexagon as a cell in a Voronoi diagram.

IDW interpolation was then used to directly predict the expected local facet surface area at each SP directly from the calculated facet areas of the CLP. On average, the expected facet surface areas were close to the interpolated hexagon areas, and in all cases, they differed by less than 15%. Given that the expected facet area was interpolated from a single parameter rather than from the 6 2D coordinates that are used for hexagon interpolation, we expected that the area of the former would be more accurate. Hence, we applied a correction to the interpolated hexagon at each SP by scaling it such that its area equaled the expected facet area. This also adjusted the distance to the adjacent facet centres, and the average of all six

distances was used as the local $D$ of a given SP. Dividing the eye's total surface area by the averaged expected facet area of all SPs indicated the total facet number for each compound eye.

## Corneal projection

For a given SP, we selected the corneal voxels that were closest to each of the six adjacent facet centres determined from hexagon interpolation. The NV at each facet centre was then calculated, and the angle between the NV of the SP and each adjacent NV indicated the divergence angle between the corneal orientation of those facets. These six angles provide two measurements for each of the x-y-z axes of an individual facet (**Land and Eckert, 1985**). The average of the six angles for a SP indicated the local IO angle ($\Delta\Phi$), whereas the average of dividing $D$ at each SP by the associated $\Delta\Phi$ (in radians) indicated the local radius of curvature ($R$, **Figure 2B**). Note that when calculated from corneal anatomy, $D$ and $R$ are typically measured first and the IF angle is then obtained by calculating $\Delta\Phi = D/R$ (**Schwarz et al., 2011**; **Bergman and Rutowski, 2016**; **Dyer et al., 2016**), whereas we obtain $\Delta\Phi$ and $D$ first and calculated $R = D/\Delta\Phi$. The reason for using a different procedure is that we were not able to implement a satisfactory method for reliably measuring the local radius directly from the corneal surface voxels. Radius of curvature is a function of the 1st and 2nd derivatives of a surface whereas the NV is based only on the 1st derivative, and calculations of the latter were more numerically stable after the discretization of the eye into voxels.

The local cornea convergence ($C$) was determined by calculating the 2D area enclosed by the hexagon of adjacent facet centres and dividing this by the solid angle subtended by the NVs projected from those centres. The corneal projection ($CP$) was calculated by dividing the eye's total surface area by the average $C$ of all SPs: $CP = A_{total}/\overline{C}$, where $A_{total}$ is the total eye surface area.

Multiplying the IF angle of each SP by $D$ provided the local eye parameter ($P$): $P = \Delta\Phi \times D$. However, note that the eye parameter is typically calculated using the IO angle (**Snyder, 1979**), which limits the accuracy with which we calculated this parameter.

Finally, the local facet axis density ($\sigma$) projected into visual space was calculated for each SP by dividing the cornea convergence by the interpolated facet area: $\sigma = C/A_{facet}$, where $A_{facet}$ is the local facet surface area.

## Sensitivity calculation

The local optical sensitivity ($S$) of an ommatidia can be calculated as: $S = A_{facet} \times \Delta\rho^2 \times (k \times L/(1 + k \times L))$ (**Warrant and Nilsson, 1998**), where $A_{facet}$ is the facet surface area, $\Delta\rho$ is the acceptance angle of the ommatidia, $L$ is the rhabdom length, and $k$ is the absorption coefficient of arthropod photoreceptors ($0.0067~\mu m^{-1}$).

We could not measure local $\Delta\rho$ or $L$ directly in this study and approximated these by $\sqrt{\sigma^{-1}}$ (essentially assuming that the inter-ommatidial (IO) angle is equal to the IF angle) and retinal thickness, respectively. This substitution assumes that the acceptance angle is equal to the IO angle, but it is often somewhat larger (**Land, 1997**) and so our sensitivity calculations are an approximation that is likely to underestimate absolute optical sensitivity.

## Determining fields of view

We generated a set of 10,242 world points (WP) equally spaced at approximately 1° intervals across the surface of a sphere that provided a common reference frame for world-centric calculations. The WPs within each bee's CP were determined with respect to the sphere intersects of the NVs from the SPs and BPs. Each SP was associated with an IF angle, and we used nearest-neighbor interpolation to find an IF angle value for each BP from the SPs. We then tested whether each WP was within one IF angle range of the sphere intersection of any SP or BP, if so, they were included in our preliminary assignment to the set of WPs within the eye's visual field ($CP_{left}$). The visual field was assumed to cover a single contiguous region of

the visual sphere, so any unassigned WPs that were completely enclosed were also assigned to $CP_{left}$. After these assignments, the fraction of the sphere covered by $CP_{left}$ always exceeded the calculated value for the CP by approximately 5%. To resolve this discrepancy, we sequentially removed WPs from the periphery of $CP_{left}$ until the angular area covered by both visual field representations was equal. $CP_{left}$ was then mirrored to the opposite side of the visual sphere to represent the CP of the bee's right eye ($CP_{right}$). The binocular CP was defined as the intersection of $CP_{left}$ and $CP_{right}$, while the union of these sets of points defines the complete CP.

## Interpolation variables onto world points

While sampling points were equally spaced on a given eye, their projected NVs did not necessarily have equal angular spacing because the eyes' radius of curvature varied. To ensure uniform angular sampling of each variable, we used IDW interpolation (with p=4) to interpolate each locally measured variable from the BPs and SPs to the WPs based on the angular distance between their sphere intersection points. The value of variables at the BPs were again found from nearest-neighbor interpolation from the values at SPs. The method of assigning values to the BPs forced the IDW interpolator to act as a nearest-neighbor extrapolation of the SPs measurements at the periphery of the visual field. The local variables interpolated to the WPs were: facet diameter ($D$), IF angle ($\Delta\Phi$), radius of curvature ($R$), thicknesses (lens, CC, and retinal), eye parameter ($P$), sensitivity ($S$), and facet axis density ($\sigma$).

## Validation of calculations

To validate the CP calculation, we integrated the projected axis density across the WPs in the CP of a given bee. This integral predicted the total number of facet axes projected from the eye, which was compared to the total number of facets found from the area-based measurement. We considered the number of facets to be relatively accurate, but also knew that the choice of local radius influenced the NV calculation (the basis for calculating $\Delta\Phi$ and $\sigma$), and must therefore influence the total axis number. If all calculations were completed using a small local radius (50 µm), the predicted number of axes was substantially greater than the number of facets, whereas fewer axes were predicted as the local radius was increased. This increase did not result, however, in the axes number converging to the number of facets, as a large local radius (500 µm) led to the calculation of an insufficient axes number. To identify a suitable local radius for the calculations on each bee's eye, we implemented a procedure that initially used a 100 µm local radius for all NV-dependent calculations, and then iteratively adjusted that parameter and repeated all calculations until the axis number was between 95% and 100% of the facet number. The required local radius was between 100 µm and 200 µm for all bees. The axis number was chosen to underestimate the facet number slightly in order to compensate for our expectation that the latter value slightly overestimated the true number of facets on an eye (see the 'Discussion' section).

## Weighting means, distributions, and correlations

We computed mean values, relative distributions, and correlation coefficients for all local variables measured on each eye. The SPs at which all eye-centric variables are calculated are equally spaced across an eye, whereas the facet density varies across it, and simply using the values for each SP directly would misrepresent the calculation of these statistics. Therefore, we calculated weighted means, distributions, and correlations, where the weight used for each SP was the inverse of local facet surface area. This weighting represented the relative facet density of a SP and was analogous to performing facet-wise statistics (as if variables were measured for each facet on the eye individually).

## Representing world-centric variables

Visual fields and the projected topology of any variable could be displayed either directly on a sphere (*Figure 1Ei*) or as a 2D projection (*Figure 1Eii*, iii). In the latter case, we preferred to use a sinusoidal projection that preserves the representation of area on a sphere (*Figure 1Eiii*) but does not preserve the angle between different points. Given the azimuth (*a*) and elevation (*e*) of a point on the sphere, the map coordinates (*x*, *y*) of a sinusoidal projection are calculated as: $x = a \times \cos(e)$, and $y = e$.

We also calculated profiles of averaged and integrated variables across bands of azimuth or elevation. To do this, we computed the azimuth and elevation of each WP, and then calculated the mean (or integral) for all WPs a bee viewed within a specific 10° range of either azimuth (for example, $0° < \text{azimuth} < 10° \mid -90° < \text{elevation} < 90°$) or elevation (for example, $0° < \text{el.} < 10° \mid -180° < \text{az.} < 180°$). Note that each azimuthal band had the same number of points, while the number of points in an elevation band decreased towards the poles (±90° el.). Averages were calculated as weighted means for the reasons given in the preceding section, with the exception being that WPs were weighted based on facet axis density.

## Comparison to the results of other studies with 'worst case' errors

In our discussion, we compare the measurements from this study directly to the results of several others. We calculated 'worst case' errors when comparing our results to other studies on honeybees. 'Worst case' error was calculated as a percentage by selecting our measurement that was furthest from the v reported in another study, and dividing it but that studies plus one standard deviation in the opposite direction. For example, compared to the findings of *Streinzer et al., 2013*, both of our measurements for honeybee facet number (5440 and 5484) are larger than the reported values (5375 ± 143), hence, we calculate the percentage error as 5484/(5375–143)=+4.8%.

## Allometry

Applying log transformation to our data and the allometric function results in the equation $\log_{10}(Y) = \log_{10}(b) + \alpha \log_{10}(X)$, for which the parameters can be obtained using a linear regression calculation. All variables were converted to linear measurements before the logarithmic transform (by taking the square root of areas and the cube root of volumes). Calculating the linear correlation of the transformed variables also provides $R^2$ and $p$ to indicate the effectiveness of a linear fit for describing the data. Functions with non-significant ($p > 0.05$) correlations are plotted as dotted lines in our figures.

We usually used eye size as the independent parameter when measuring visual allometry as we wished to focus on investigating how, given an initial total investment in eye size, the resolution or sensitivity of an eye was improved. In other contexts, it may be desirable to use body size as the independent variable, which can be accomplished by multiplying the scaling exponent measured for a specific variable vs. eye size by the scaling exponent for bumblebee eye size vs. body size (0.45).

## Limitations of our technique

We acknowledge that our technique has several limitations that primarily relate to our inability to measure the internal dimensions of each ommatidia across the eye. To provide a calculation of sensitivity, it is necessary to determine the local rhabdom diameter and the focal length of the lens (*Warrant and Nilsson, 1998*). Rhabdoms generally could not be consistently visualized in our imaged volumes (*Figure 1A,D*); their diameter was probably close to the voxel size (1.6 μm). The focal length is an optical property of a lens and requires knowledge of its inner and outer radii, its thickness, and its refractive index to be calculated (*Born and Wolf, 1999*). Although our measurements show that lens thickness varies across each bee's eye (suggesting that the focal length is also likely to vary), the raw volumes did not have sufficient resolution to allow us to determine the outer and inner lens radii of individual facets on the

eye (note that these are not equivalent to radius of curvature calculated in relation to IF angle). Improving the resolution and contrast with which the entire eye is imaged would allow the rhabdom diameter (and length) as well as the corneal lens structure of individual ommatidia to be measured and used in optical calculations.

In addition, our descriptions of CP and IF angles are from the corneal level and assume that the optical axes of ommatidia are perpendicular to the cornea. Studies have shown, however, that the optical axes of ommatidia are often skewed from the lens normal (*Stavenga, 1979*) and extreme differences of up to 40° have been reported in the ventral rim of honeybee eyes (*Baumgärtner, 1928*). In such cases, the visual field can be expanded at the cost of reduced visual resolution and optical sensitivity (the effective lens diameter is reduced and corneal reflections increase, although these may be offset by an increased acceptance angle). We could visualize the CC across parts of some compound eyes (*Figure 1B,C*), but were unable to reliably segment them individually, and thus we were unable to incorporate their skewness into our analysis. Hence, we strictly use the terms CP and IF angle (rather than FOV and IO angle); we believe that these measurements provide a suitable basis for comparison between subjects that are likely to have similar patterns of CC offsets (such as between bees), although this assumption may not hold if comparisons are made between species with particularly different eye shapes (such as between the oval eyes of bees and the spherical eyes of butterflies). Note that we assumed that the IO angle was equal to the IF angle that we computed when calculating optical sensitivity and eye parameter, which also limits the accuracy with which we can report those values.

An additional limitation of this method is that we calculate an average IF angle at each sampling location on a compound eye and do not decompose this into partial angles for the facet's vertical and horizontal axes (*Figure 1C*) (*Stavenga, 1979*). The resolution of many compound eyes is asymmetric such that the resolution in the vertical and horizontal directions differs (*Land, 1997*). Sampling asymmetry occurs because either the hexagonal lattice of ommatidia is elongated or the eye's radius of curvature varies directionally. Indeed, the oval eyes of bees are generally known to have higher vertical resolution than horizontal resolution in their frontal visual field (*Seidl, 1982*; *Spaethe and Chittka, 2003*). In this study, we chose to average IF angles because we often found that their peripheral facets were packed irregularly and did not provide an obvious choice for facet axis, or that the facet axes were misaligned from the world axes (also noted by *Seidl and Kaiser, 1981*). Both factors complicate the calculation of partial angles across the entire visual field in a manner that would be analogous to the decomposed angles presented elsewhere. The average angle can, however, be calculated from the vertical and horizontal IO angles reported in other studies and compared directly to the values in our study.

Finally, we noticed that the relative accuracy of our method of calculating IF angles declined on the flattest compound eye areas. This was because the discretization of an imaged eye surface into voxels made areas with a large radius of curvature indistinguishable from flat planes. We determined IF angles by fitting polynomial surfaces to the voxels from small areas on the surface and then calculating the angle between the NVs from points on these functions. If the function was fit to a flat plane, then the NVs from all points across it would lie in parallel (indicating a misleading IF angle of 0°). To prevent such inaccuracies, we implemented a heuristic that enlarged the local radius used to select the voxels used when fitting functions until it selected a curved set of points. Nonetheless, it is apparent that the largest relative errors for this method of calculating IF angle are likely to occur in the highest resolution areas of an eye, as these are also likely to be the flattest.

