## [Decision Letter]

[Editors’ note: the authors were asked to provide a plan for revisions before the editors issued a final decision. What follows is the editors’ letter requesting such plan.]

Thank you for sending your article entitled "Bumblebee visual allometry results in locally improved resolution and globally improved sensitivity" for peer review at *eLife*. Your article is being evaluated Joshua Gold as the Senior Editor, a Reviewing Editor, and three reviewers. The reviewers have opted to remain anonymous.

Given the list of essential revisions, including the possibility of new experiments, the editors and reviewers invite you to respond within the next two weeks with an action plan and timetable for the completion of the additional work. We plan to share your responses with the reviewers and then issue a binding recommendation.

Summary:

This paper uses a relatively new technique, X-ray micro-computed-tomography to examine how topologies of several key optical parameters scale relative to overall eye volume across 6 bumblebees. Although several groups have used volumetric reconstructions with microCT, the approach here is to model the ommatidial surface and use the segments of the various portions of the eye along with the normals from the surface to extract the visual parameters. Further, the authors measured the same parameters in two honeybees to compare previously validated measurements. Compound eye visual function is limited by structural properties (field of view, inter-ommatidial angle, facet number…). These parameters scale with eye size and can vary across eye regions.

Using this projective analysis, the study shows that, in general, larger eyes afford greater FOV, greater binocular overlap, a thicker retina, and larger facets that, for the most part, contribute to increases in sensitivity. The study further demonstrated regional differences in these results. Increases in FOV for large bees are mostly in the regions of binocular overlap. Decreases in IO angle (implying increases in acuity) are primarily near the central visual field (~15 degrees azimuth). Increases in facet diameter (implying increases in sensitivity) are nearly homogeneous across the whole eye while increases in retinal thickness are more dramatic in the dorsal visual field, suggesting greater increases in sensitivity in the dorsal visual field. Finally, using Snyder's eye parameter, the authors show how large eyes invest in acuity versus sensitivity across the eye: acuity increases primarily in a small portion of the central and dorsal frontal visual field while sensitivity increases everywhere else, with the most dramatic increases in the lateral region and the regions of binocular overlap.

Essential revisions:

1) The most serious concern was about interpretations of the results in terms of visual field properties and light sensitivity, given the lack of validation of the interommatidial angles (which, it was noted, is traditionally done on a living eye). As you certainly know, getting these angles correct is very important because the physiological function is complex, and apparently in bees the photoreceptor is not always perfectly on axis with the rest of the ommatidial morphology. The reviewers noted that you acknowledge these issues at various places in the paper; e.g. that there may be errors as large as 30% to 60%. However, you also make claims that your study demonstrates that the new microCT-based method could be used "for reconstructing the visual world of insects with unprecedented detail." Given the lack of validation of the interommatidial angles and potentially large errors, the levels of confidence expressed in these kinds of statements about visual field properties and light sensitivity were considered to be overstatements given the level empirical support.

Therefore, to continue to make such claims, it would be necessary for you to collect more data and appropriately validate your reported measured of the interommatidial angles. However, absent those data, it would be necessary to clearly state all assumptions made regarding parameters involved in modeling eye performance, and to substantially revise how you describe both the nature and reliability of the claims made throughout the manuscript (the reviewers agreed that the paper was interesting even without strong claims about visual field properties and light sensitivity).

2) More generally, there was agreement that the manuscript needs a very major re-write to be appropriate for a general readership. Several specific suggestions are as follows:

a) The Introduction is too long and goes into too much detail on some basics of insect vision. It would be better to stick to a select few main ideas that would help a reader assess the importance of the study (for example, what does the energetic cost of eye development have to do with this current paper?).

b) Materials and methods section – can you provide a diagram illustrating all measurements? Or add ITW to Figure 1 or Figure 2, which could probably be combined. Elevation and azimuth or terminology to describe your transects should be labeled on this figure. Some of the methods are written with too much granular detail, and it is not always clear what the purpose of the method is as it is revealed at the end of the paragraph.

c) Figure 3 – the green allometric equation is hard to read. It is not clear that the legend for *Apis/Bombus* in panel A applies to all figures. Can you make it more prominent? Or put all legends together? Why is it "Corneal IO angle"? Is there another IO angle in your paper? Just define it in your methods and then simplify your labels to make it easier to follow.

d) The presentation of parameters should be grouped – for example, Facet Diameter and IO angle are usually the most important ommatidial parameters that dictate visual function (and retinal thickness as a proxy for rhabdom length), eye area and volume as overall parameters.

3) Aspects of the analysis are unclear. For example, in allometric equations, what is the parameter that is being fit to the allometric equation? All parameters are measured all around the eye, so is it the global average per eye? If so, isnʻt there a better way to do this analysis? Why not apply some multivariate morphometrics such as geometric morphometrics? In the subsection “Allometry”, the power function is linearized and then a linear correlation is computed? To test for model fit? This may not be the best way, you can check whether a linear model or a quadratic, for example, is a better fit (in general you have to do model comparison to decide if it is a better fit).

[Editors’ note: formal revisions were requested, following approval of the authors’ plan of action.]

---

## [Author Response]

[Editors’ note: what follows is the authors’ plan to address the revisions.]

Essential revisions:1) The most serious concern was about interpretations of the results in terms of visual field properties and light sensitivity, given the lack of validation of the interommatidial angles (which, it was noted, is traditionally done on a living eye). As you certainly know, getting these angles correct is very important because the physiological function is complex, and apparently in bees the photoreceptor is not always perfectly on axis with the rest of the ommatidial morphology. The reviewers noted that you acknowledge these issues at various places in the paper; e.g. that there may be errors as large as 30% to 60%. However, you also make claims that your study demonstrates that the new microCT-based method could be used "for reconstructing the visual world of insects with unprecedented detail." Given the lack of validation of the interommatidial angles and potentially large errors, the levels of confidence expressed in these kinds of statements about visual field properties and light sensitivity were considered to be overstatements given the level empirical support.Therefore, to continue to make such claims, it would be necessary for you to collect more data and appropriately validate your reported measured of the interommatidial angles. However, absent those data, it would be necessary to clearly state all assumptions made regarding parameters involved in modeling eye performance, and to substantially revise how you describe both the nature and reliability of the claims made throughout the manuscript (the reviewers agreed that the paper was interesting even without strong claims about visual field properties and light sensitivity).

Yes, as stated, we do not have pseudopupil measurements of interommatidial angles from live bees, which is the classic ‘gold standard’ against which our calculations based upon the shape of the insects cornea could be compared. We chose not to perform these because bees have black eyes and so pseudopupil measurements must be made using antidromic illumination, which is a difficult and time-consuming measurement technique to use (as far as we know honeybees are the only species where this technique has been used to map a complete visual field – see Sidel, 1982), and it would also not be practical to perform on the same samples we analyzed using CT. Indeed, a motivation for developing our CT based technique was to provide a higher throughput method for performing comparative studies on the visual fields of a wide range of bee species, which would be impossible using the pseudopupil method.

- To address this, we firstly plan to change the terminology used in the paper to refer to ‘inter-facet angles’ and ‘corneal projection’ to avoid confusion with the classic usages of ‘inter-ommatidial angle’ and ‘field of view’ respectively. We believe that our values should still be compared to classical measurements in the discussion (such as comparing ‘inter-facet angle’ to ‘inter-ommatidial angle’) as this provides a standard against which the accuracy of our methods measurement of visually relevant parameters from CT scans can be determined. We will also clarify in the discussion that while the percentage difference between our findings and classical measurements can be relatively large (30-60%), in many cases this corresponds to relatively small absolute differences of less than 0.5 degrees (see Table S2 in Supplementary file 1).

- Secondly, we agree that the assumptions about using our corneal values in place of the classical measurements needs to be stated more clearly in the main body of the manuscript, particularly with respect to when they are used to calculate values such as Land’s sensitivity equation or Snyder’s eye parameter. We will also note these assumptions when discussing our findings from employing these equations, although we think that calculating them still provides useful data about bee vision (such as the likelihood of high acuity regions of vision, and regional variation in sensitivity, which has been noted by studies using other methods).

- Finally, we agree that we have somewhat overstated the accuracy with which we can currently reconstruct an insect’s visual world using this analysis technique – we will modify statements about this accordingly.

2) More generally, there was agreement that the manuscript needs a very major re-write to be appropriate for a general readership. Several specific suggestions are as follows:a) The Introduction is too long and goes into too much detail on some basics of insect vision. It would be better to stick to a select few main ideas that would help a reader assess the importance of the study (for example, what does the energetic cost of eye development have to do with this current paper?).

We agree that the Introduction is too lengthy and should be shortened. We will remove the section describing energetics and most of the material related to vertebrate eyes. We will also condense the material related to eye scaling, and we think that this will allow us to replace the first six paragraphs of the

Introduction with three, more focused, paragraphs.b) Materials and methods section – can you provide a diagram illustrating all measurements? Or add ITW to Figure 1 or Figure 2, which could probably be combined. Elevation and azimuth or terminology to describe your transects should be labeled on this figure. Some of the methods are written with too much granular detail, and it is not always clear what the purpose of the method is as it is revealed at the end of the paragraph.

- Figures 1 and 2 will be combined for brevity.

- ITW was measured on the thorax of the bees and we will illustrate this with a pictogram in the combined figure (which currently show diagrams of the bee head and eyes).

- We will include pictograms to indicate how our transects (actually either sums or averages across azimuth and elevation) are measured in the figures where they are presented (Figures 4, 5 and 6 and several figure supplements).

We will edit the Materials and methods section so that the purpose of each step is clear, and shift some more of the implementation details to the supplementary material.c) Figure 3 – the green allometric equation is hard to read. It is not clear that the legend for Apis/Bombus in panel A applies to all figures. Can you make it more prominent? Or put all legends together? Why is it "Corneal IO angle"? Is there another IO angle in your paper? Just define it in your methods and then simplify your labels to make it easier to follow.

- We will darken the green symbols/line and make the legends more cohesive for readability.

- We currently emphasise corneal IO/corneal FOV in our text and figures in an attempt to distinguish them from their classical definitions. We already describe renaming these variables to address comment 1, which we will remove.

this confusion.d) The presentation of parameters should be grouped – for example, Facet Diameter and IO angle are usually the most important ommatidial parameters that dictate visual function (and retinal thickness as a proxy for rhabdom length), eye area and volume as overall parameters.Agreed, we will shift the angle and diameter histograms from Figure 3D and E to place above the topologies into Figure 6A and B (such that angle and diameter are grouped like the rhabdom length in Figure 5).3) Aspects of the analysis are unclear. For example, in allometric equations, what is the parameter that is being fit to the allometric equation? All parameters are measured all around the eye, so is it the global average per eye? If so, isnʻt there a better way to do this analysis? Why not apply some multivariate morphometrics such as geometric morphometrics? Subsection “Allometry”. The power function is linearized and then a linear correlation is computed? To test for model fit? This may not be the best way, you can check whether a linear model or a quadratic, for example, is a better fit (in general you have to do model comparison to decide if it is a better fit).

We will describe our analysis as clearly as possible in the relevant method text, figure legends and supplementary material. The allometry equation is indeed fit to the global average per eye (and we will state this more clearly in the text) – using morphometric techniques to look at the key modes of variation (for one or more variables across the visual field) would be possible in principle. However, we did not pursue this for two reasons: firstly, our sample size is relatively small (six *Bombus* individuals), while we have values for measured variables from a large number of points on the visual field, which is likely to result in relatively unreliable results from the dimensional reduction approaches used in geometric morphometrics. Secondly, we felt that the paper already introduces large number of new concepts for insect vision and we did not feel that it was appropriate to introduce and describe another (we do recognise the value of morphometric techniques and are exploring their use in a further study that compares eye differences between *Bombus* species). We did, however, attempt to examine how scaling varies across eyes by plotting the local scaling parameter for all measured variables (Figure 7), and we think that continuing to use the current maps of local scaling parameters is preferable to also incorporating a morphometric approach. We used a power function to describe allometry (which was fit as a linear correlation after linearising the data) as this has been a standard approach to describe trait allometry since it was introduced by Huxely and Teisseir (1936). While other models may provide a better fit to the data, a power function has been widely used in many studies which allows the coefficients to be compared between species/groups (as we do in the Discussion). We therefore believe it is appropriate and most useful for the readers to continue using a power function to model allometry.

[Editors’ notes: the authors’ response after being formally invited to submit a revised submission follows.]

Essential revisions:1) The most serious concern was about interpretations of the results in terms of visual field properties and light sensitivity, given the lack of validation of the interommatidial angles (which, it was noted, is traditionally done on a living eye). As you certainly know, getting these angles correct is very important because the physiological function is complex, and apparently in bees the photoreceptor is not always perfectly on axis with the rest of the ommatidial morphology. The reviewers noted that you acknowledge these issues at various places in the paper; e.g. that there may be errors as large as 30% to 60%. However, you also make claims that your study demonstrates that the new microCT-based method could be used "for reconstructing the visual world of insects with unprecedented detail." Given the lack of validation of the interommatidial angles and potentially large errors, the levels of confidence expressed in these kinds of statements about visual field properties and light sensitivity were considered to be overstatements given the level empirical support.Therefore, to continue to make such claims, it would be necessary for you to collect more data and appropriately validate your reported measured of the interommatidial angles. However, absent those data, it would be necessary to clearly state all assumptions made regarding parameters involved in modeling eye performance, and to substantially revise how you describe both the nature and reliability of the claims made throughout the manuscript (the reviewers agreed that the paper was interesting even without strong claims about visual field properties and light sensitivity).

We completed the changes to this section as planned, by:

– Introducing the terminology Corneal Projection (CP) and Inter-facet (IF) angle;

– Noting in the Materials and methods section, figure legends, Results, and supplementary material that our calculations of the eye parameter and sensitivity were based on assumptions about the Inter-facet angle equaling the Inter-ommatidial angle. Now that we have renamed our variables, the comparison of our IF angles to reported IO angles is clearer;

– Moderating our description about the reliability of our visual measurement technique.

2) More generally, there was agreement that the manuscript needs a very major re-write to be appropriate for a general readership. Several specific suggestions are as follows:a) The Introduction is too long and goes into too much detail on some basics of insect vision. It would be better to stick to a select few main ideas that would help a reader assess the importance of the study (for example, what does the energetic cost of eye development have to do with this current paper?).

We have shortened the Introduction to about two thirds of its original length and edited it so our ideas related to visual field allometry are more clearly expressed.

b) Materials and methods section – can you provide a diagram illustrating all measurements? Or add ITW to Figure 1 or Figure 2, which could probably be combined. Elevation and azimuth or terminology to describe your transects should be labeled on this figure. Some of the methods are written with too much granular detail, and it is not always clear what the purpose of the method is as it is revealed at the end of the paragraph.

We have combined the first two figures and added the ITW measurement as requested. We also added pictograms to illustrate the meaning of the azimuth/elevation profiles in several figures, and we hope that their meaning is now clear. We have shifted approximately half of the Materials and methods section to supplementary material.

c) Figure 3 – the green allometric equation is hard to read. It is not clear that the legend for Apis/Bombus in panel A applies to all figures. Can you make it more prominent? Or put all legends together? Why is it "Corneal IO angle"? Is there another IO angle in your paper? Just define it in your methods and then simplify your labels to make it easier to follow.These points were addressed as suggested.d) The presentation of parameters should be grouped – for example, Facet Diameter and IO angle are usually the most important ommatidial parameters that dictate visual function (and retinal thickness as a proxy for rhabdom length), eye area and volume as overall parameters.

These points were addressed as suggested.

3) Aspects of the analysis are unclear. For example, in allometric equations, what is the parameter that is being fit to the allometric equation? All parameters are measured all around the eye, so is it the global average per eye? If so, isnʻt there a better way to do this analysis? Why not apply some multivariate morphometrics such as geometric morphometrics? Subsection “Allometry”. The power function is linearized and then a linear correlation is computed? To test for model fit? This may not be the best way, you can check whether a linear model or a quadratic, for example, is a better fit (in general you have to do model comparison to decide if itʻs a better fit).

We have clarified how the allometric function was indeed fitted to the means in the Materials and methods section. As discussed in our initial plan, we did not feel it was appropriate to start using morphometric analysis or testing various model fits. However, we do include a suggestion for the future use of, and references to, morphometric techniques in the Discussion section.